# Towards fully ab initio simulation of atmospheric aerosol nucleation

Shuai Jiang [1] ✉, Yi-Rong Liu [1], Teng Huang [2], Ya-Juan Feng [1], Chun-Yu Wang [1], Zhong-Quan Wang[2], Bin-Jing Ge [1], Quan-Sheng Liu [1], Wei-Ran Guang [1] & Wei Huang [1,2,3]

Atmospheric aerosol nucleation contributes to approximately half of the worldwide cloud condensation nuclei. Despite the importance of climate, detailed nucleation mechanisms are still poorly understood. Understanding aerosol nucleation dynamics is hindered by the nonreactivity of force fields (FFs) and high computational costs due to the rare event nature of aerosol nucleation. Developing reactive FFs for nucleation systems is even more challenging than developing covalently bonded materials because of the wide size range and high dimensional characteristics of noncovalent hydrogen bonding bridging clusters. Here, we propose a general workflow that is also applicable to other systems to train an accurate reactive FF based on a deep neural network (DNN) and further bridge DNN-FF-based molecular dynamics (MD) with a cluster kinetics model based on Poisson distributions of reactive events to overcome the high computational costs of direct MD. We found that previously reported acid-base formation rates tend to be significantly underestimated, especially in polluted environments, emphasizing that acid-base nucleation observed in multiple environments should be revisited.

The theoretical understanding of the nucleation mechanism largely relies on classical nucleation theory (CNT)[1], originally proposed in 1935, which gives a general mind map for nucleation thermodynamics and kinetics[2] even though the capillary assumption has been extensively criticized[3]. The theoretical model Atmospheric Cluster Dynamics Code (ACDC), which emerged[4] in 2011 and was subsequently broadly employed[5–10], surmounts the drawbacks of CNT through coupled quantum chemical thermodynamics[11] with birth–death equations[2]. In the framework of ACDC, collision rate constants and evaporation rates are the two most critical parameters, determining the accuracy of the prediction of macroparameters such as cluster concentrations and formation rates that can be directly determined with experiments for comparison[5]. Evaporation rates, derived from detailed balance and ab initio thermodynamics, can be very accurately obtained with sophisticated quantum chemical calculations[12]. However, collision rate constants, derived from a simple hard-sphere

collision model, are still very rough, and the accuracy is far from that of ab initio-based evaporation rates. Moreover, determining accurate collision rate constants is extremely important, especially for collision-controlled systems such as sulfuric acid–dimethylamine systems, as evaporation rates are close to zero[13]. Pioneering work[14] investigated the collisions between sulfuric acid monomers; however, the force field (FF) utilized lacks reactivity, and the computational costs of extending the method to more collisions among molecules and/or clusters are enormous. Therefore, a highly accurate and inexpensive reactive FF for flexible nucleation clusters is urgently needed to simulate nucleation processes with full ab initio accuracy.

Here, we propose a general workflow to drive the aerosol nucleation simulation toward becoming fully ab initio. In the workflow, comprehensive data sets are first prepared through metadynamics coupled with active learning techniques. Then, a deep neural network-based force field (DNN-FF) is trained so that robust nucleation

[1]School of Information Science and Technology, University of Science and Technology of China, Hefei, Anhui 230026, China. [2]Laboratory of Atmospheric Physico-Chemistry, Anhui Institute of Optics & Fine Mechanics, Chinese Academy of Sciences, Hefei, Anhui 230031, China. [3]Center for Excellent in Urban Atmospheric Environment, Institute of Urban Environment, Chinese Academy of Sciences, Xiamen, Fujian 361021, China. ✉e-mail: shuaijiang@ustc.edu.cn

molecular dynamics (MD) simulations can be performed to derive the collision rate constants based on the Poisson distribution. Then, static quantum chemical thermodynamics-based evaporation rates are coupled with DNN-FF-based MD-derived collision rate constants into a cluster dynamics model to provide ab initio kinetics for simulating atmospheric aerosol nucleation.

## Results

### A general workflow for fully ab initio simulation of aerosol nucleation

The key modules in the workflow are shown in Fig. 1. The details in each module can be found in the Methods section, so here, the major points regarding the significance and correlation for each module are given. The initial data set is first prepared by metadynamics sampling in addition to subsequent screening and labeling. The screening is made by farthest point sampling (FPS), while the force and energy labeling is done by density function theory (DFT). Then, an active learning strategy with two force thresholds is utilized to supplement the initial data set to form the final data set to obtain the final force field. In each active learning iteration, DNN-FF-based MD based on the previously active learning iterations selected data set and metadynamics prepared data set is conducted. Then, inaccurate structures satisfying the threshold range are selected for labeling and added to the data set for

the next iteration. Therefore, after finalizing the data set, multiple DNN-FF-based one nanosecond MD simulations can be performed. Finally, based on the Poisson distribution, collision rate constants are derived and combined with static quantum chemistry-based evaporation rates to obtain macroparameters such as the formation rate by a cluster kinetics model. The cluster size sampled by metadynamics is based on the cluster stability characteristic of acid-base clusters being mostly stable when the difference between an acid number and the base number is less than or equal to one[5]. Active learning not only supplements the structures for metadynamics sample size but also points to the cluster compositions with high evaporation rates, e.g., $(DMA)_4$, the cluster being composed of four dimethylamine molecules, as we can see from the active learning data set in Fig. 1c, which can normally be ignored through sampling under predefined cluster compositions. Notably, we will use $(SA)_m(DMA)_n$ to represent the cluster composed of m sulfuric acid molecules and n dimethylamine molecules. Collision rate constants, derived from MD simulations based on Poisson distribution reaction events (Fig. 1d), are essentially independent of cluster concentrations, making high-concentration MD simulations valuable for further cluster kinetics.

Due to the interpolative nature of DNNs, high accuracy could be maintained for clusters up to $(SA)_{10}(DMA)_{10}$. The accuracy for clusters beyond $(SA)_{10}(DMA)_{10}$ is unknown, but we expect a further decrease in

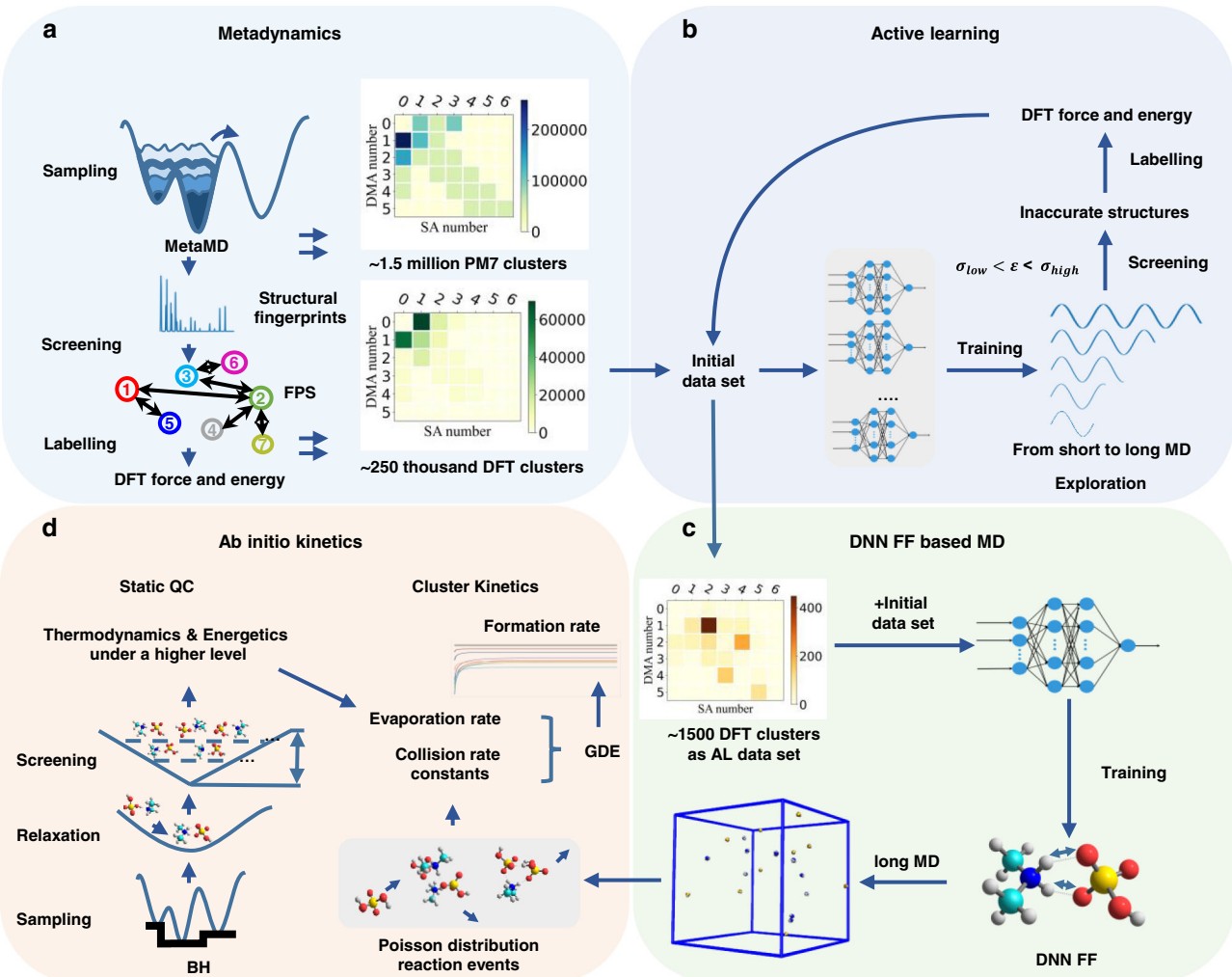

**Fig. 1 | A general workflow towards fully ab initio simulation of atmospheric aerosol nucleation.** It includes the steps to prepare the data set for training a deep neural network-based force field (DNN-FF), to apply DNN-FF by molecular dynamics (MD), to derive collision rate constants from MD, and to couple collision rate constants with cluster kinetics model for studying atmospheric aerosol nucleation. **a**, **b** show the metadynamics and active learning techniques used to prepare a data set for the deep neural network, respectively. **c** DNN-FF-driven MD. **d** Cluster kinetics simulation based on MD-derived collision rate constants and static quantum chemistry (QC) calculation-derived evaporation rates.

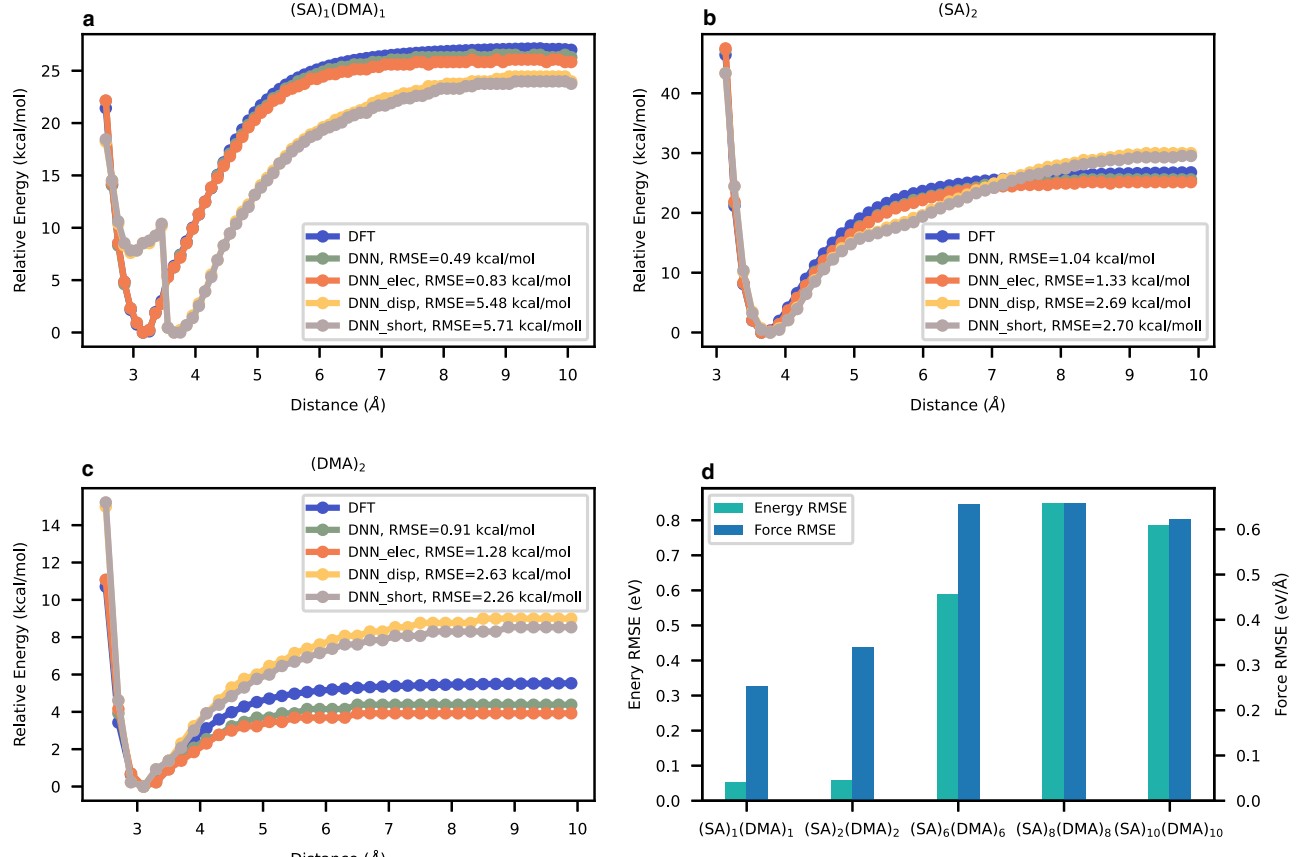

**Fig. 2 | Deep neural network-based force field (DNN-FF) benchmark. a–c** show the dimer detachment curves for $(SA)_1(DMA)_1$, $(SA)_2$, and $(DMA)_2$, respectively, where SA and DMA represent sulfuric acid and dimethylamine molecules, respectively. The relative energy is the isomer energy minus the energy of the most stable isomer. DNN, DNN_elec, DNN_disp, and DNN_short represent the model with electrostatics and dispersions, the model with electrostatics and without dispersions, the model without electrostatics and with dispersions, and the model without electrostatics and dispersions, respectively. **d** Energy and force root mean squared error (RMSE) values in the interpolation and extrapolation regimes of the test set. Source data are provided as a Source Data file.

accuracy. The ultimate goal of simulating atmospheric aerosol nucleation is to conduct MD under ambient or laboratory conditions, where the cluster size can easily go beyond the interpolation regime, so how to apply the DNN model with size-limited accuracy to atmospheric nucleation becomes a problem that needs to be addressed. We bridge the gap between microparameters and macroparameters by embedding MD-derived rate constants based on the Poisson distribution into a cluster kinetics model. These DNN-FF-based MD-derived constants coupled with static quantum chemistry (QC)-derived evaporation rates effectively drive the aerosol simulation towards full ab initio calculations.

### The benchmark of the DNN-FF

The dimer detachment curves in Fig. 2a–c provide a basic picture of the performance of DNN-FFs significantly affected by long-range interactions. Generally, DNN-FF with long-range interactions performs very well on the investigated dimer systems, with the root mean squared error (RMSE) of the relative energy being close to 1 kcal/mol, the so-called chemical accuracy. Adding electrostatics and dispersion corrections into the short-range DNN model not only decreases the RMSE but also improves the curve smoothness. From the RMSE, electrostatic interactions are clearly more important than dispersion. The maximum cluster size within the training set is $(SA)_5(DMA)_6$, so the energy and force RMSE values in the extrapolation regime are larger than those in the interpolation regime, as expected. Even in the extrapolation regime, the DNN model still yields an encouraging accuracy close to that of a recently reported combustion reaction DNN[15]. In summary, the DNN model's superior performance in energy

and force descriptions, in addition to its distinguished size scalability, lays a solid foundation for robust nucleation MD simulations.

### Structural and energetic characteristics from DNN-FF-based MD

With the robust size scalability of the DNN model, nanosecond-scale MD simulations for cluster collision and evaporation can be performed, and a representative snapshot is shown in Fig. 3a. Furthermore, isolated clusters can be singled out to gain insights into their structural evolution. Here, $(SA)_6(DMA)_6$ is chosen since it is the largest cluster with the same number of acids and bases observed in the DNN-FF-based MD and is very close to the lowest experimentally detectable cluster size (~1.7 nm)[5]. For the most stable isomer (Fig. 3b), the sulfuric acid molecules are hydrogen bonded with each other, forming a shell with the cluster center of mass (COM) inside, while all dimethylamine molecules are protonated. The structural similarity can be seen through the closely connected points in the energy basin (Fig. 3d). In addition, here, the $(SA)_6(DMA)_6$ cluster emerges from the collision of $(SA)_4(DMA)_5$ and $(SA)_2(DMA)_1$; the high-energy isomers during a collision and subsequent rearrangement can also be seen in Fig. 3d (semitransparent red points in the upper left corner). Interestingly, none of the proton-transferred nitrogen-oxygen bonds break during the simulation (Supplementary Fig. 5), indicating quite strong bonding from proton transfer. Comparatively, we see that one of the two protons initially covalently bonded with the oxygen atom in the sulfuric acid molecule transfers to one dimethylamine molecule, while the other proton moves from one sulfuric acid molecule to another (Supplementary Figs. 4, 6). After the collision, the molecules in the cluster rearrange, finally making the proton number within sulfuric

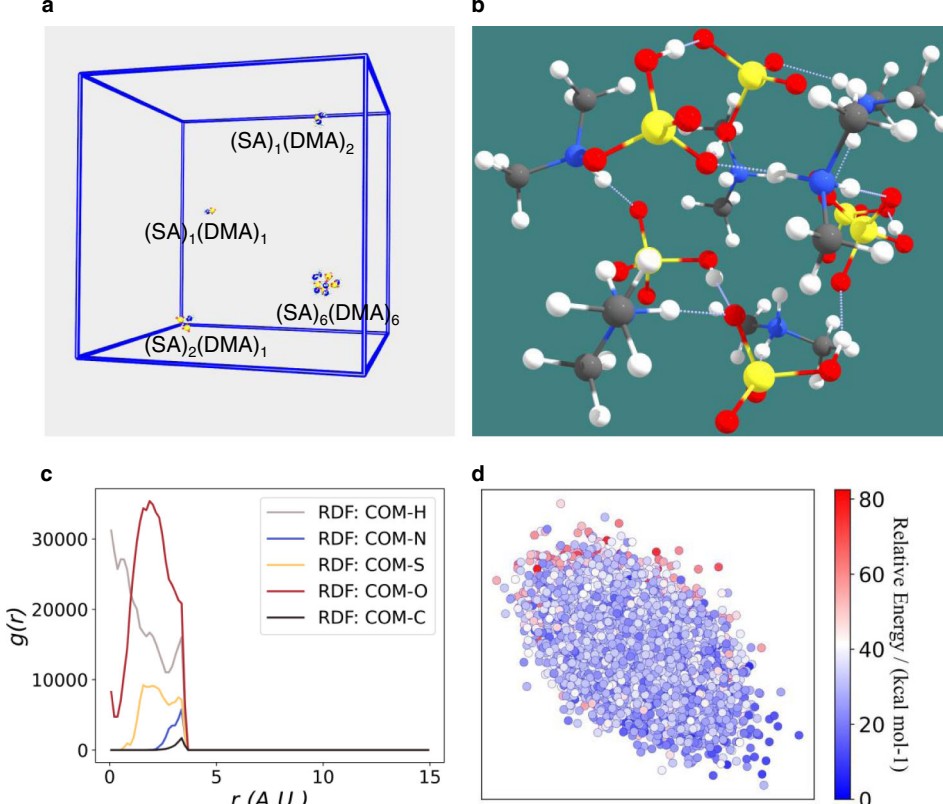

**Fig. 3 | Structural distribution for $(SA)_6(DMA)_6$ isomers derived from deep neural network-based force field (DNN-FF)-based molecular dynamics (MD).** **a** shows a snapshot of MD at 1 ns. The cyan, white, red, blue, and yellow circles represent C, H, O, N, and S atoms, respectively. The N and S atom radii are increased threefold for clarity. **b** The most stable isomer in the trajectory. **c** Radial distribution function (RDF) between the cluster center of mass (COM) and the five elements. **d** Kernel principal component analysis (KPCA)[58] maps of isomers using a global Smooth Overlap of Atomic Positions (SOAP)[59] kernel. Source data are provided as a Source Data file.

acid molecule one (Supplementary Fig. 6). From the above analysis, essential structural insights can be obtained by collecting clusters with the same composition from the MD trajectory.

## Aerosol nucleation kinetics

DNN-FF-driven MD simulation of molecular cluster collisions and evaporations follow a Poisson distribution[16], so the so-called collision enhancement factor (CEF) can be derived. This term is the quotient of MD-derived collision rate constants divided by hard-sphere collision model-derived collision rate constants. The CEF is typically more than one due to long-range intermolecular forces[14]. However, for the collision between dimethylamine monomers, the CEF is below one at 300 K, mainly because of the intermolecular repulsion of dimethylamine molecules (Fig. 4a). From Supplementary Table 1, for the same reaction, the CEF at 278 K is slightly larger than the CEF at 300 K. Replacing hard-sphere collision rate constants with MD-derived constants in a cluster kinetics model paves the way for fully ab initio simulation of aerosol nucleation. The derived formation rates divided by those based on hard-sphere collision rate constants give the formation rate enhancement factor (FREF) (Fig. 4c, d). In representative clean (Fig. 4c) and polluted (Fig. 4d) environments, FREF has a strong negative correlation with the DMA concentration and a weak negative correlation with the SA concentration. Under the typical SA and DMA concentrations with different CS values, FREF in the polluted environment is much larger than FREF in the clean environment (yellow points in Fig. 4c, d). This indicates that aerosol particle formation in a polluted environment is much more underestimated than that in a clean environment. Generally, here, FREF ranges from one to several hundreds; however, this is the lower bound value, as there are still many collision rate constants that need to be replaced. Therefore, obtaining larger formation rates than expected before challenges the idea that SA-DMA nucleation is collision-limited with zero evaporation rates[17], providing the alternative scenario that collision rate constants are underestimated when evaporation rates are low but not zero. Further studies regarding this topic are definitely needed when fully ab initio kinetics are available in the future.

## Discussion

Currently, the coupling between machine learning (ML) and chemistry is on the rise, so training a DNN model with good performance on training and test data sets is becoming increasingly routine[18–26]. However, training a DNN model with good size scalability and applicability to reactive MD simulations, especially for flexible molecules, as in this case, is still very challenging[27]. Here, robust MD simulations prove the high quality of data sets given by metadynamics and active learning, as well as the excellent performance of descriptors combined with the neural network framework and parameters. The workflow we propose here works for strong acid and strong base nucleation systems, which are the most significant systems in the aerosol nucleation field due to their strong nucleation ability. However, for systems with apparent barriers, such as the sulfuric acid-ammonia system, it is probably vital to utilize the strategy[28] to introduce transition state configurations to obtain a uniformly accurate model.

Future work can be first conducted on how to produce a compact data set, probably heavily relying on active learning. Another aspect that should be investigated is improving the DNN model accuracy, possibly through transfer learning[29] or Δ-learning[30]. For nucleation applications, more diverse box size and initial monomer spatial

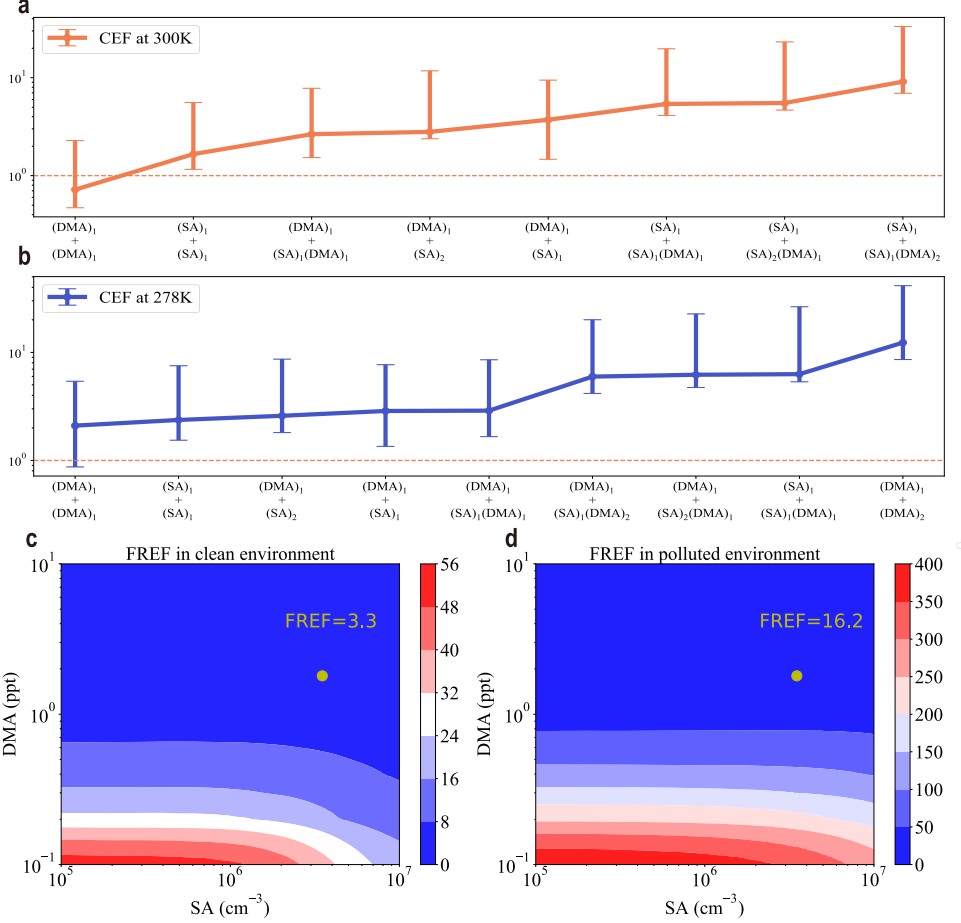

**Fig. 4 | Collision and formation rate enhancement factor (CEF and FREF).**
**a**, **b** show the collision dependence of the CEF at 300 and 278 K, respectively. The reaction list can be found in Supplementary Table 1. The error bars shown in **a**, **b**, which represent the upper and lower bound of CEF, result from a 95% confidence level of Poisson-distributed reactions. Events **c**, **d** give the FREFs for representative clean (T = 278 K, CS = 2.6 × 10⁻³ s⁻¹) and polluted environments (T = 278 K, CS = 2.7 × 10⁻² s⁻¹), respectively, where T and CS are the temperature and the condensation sink coefficient, respectively. A temperature of 300 K is chosen to resemble the conditions of standard temperature and pressure (STP), while 278 K represents typical spring-time conditions at the boreal forest site in Hyytiälä[60], the flagship observatory in the new particle formation field and winter-time conditions for observation in Beijing[61]. Yellow points in **c**, **d** show FREFs under typical clean and polluted environments, respectively, where [SA] = 3.5 × 10⁶ cm⁻³ and [DMA] = 1.8 ppt. Source data are provided as a Source Data file.

distribution MD simulations need to be conducted to lower the uncertainties[16] to estimate collision rate constants. In addition, a larger box-size simulation with more molecules inside is needed to include more types of collisions so that the simulation can be fully ab initio. Afterward, many ab initio-derived collision rates could tentatively be predicted purely by molecular physical chemistry properties to reduce simulation time costs.

The complexity and variety of nucleation precursors in the ambient environment, especially in polluted environments, necessitate new theoretical methods in addition to static QC calculations to unravel the complicated associated mechanisms. We believe the workflow proposed here, with the introduction of the DNN-FF and the bridging between MD-derived rate constants with cluster kinetics, paves the way toward the full ab initio simulation of atmospheric aerosol nucleation. The highly accurate formation rate derived here can be further parametrized into a climate model to improve climate prediction on global and local scales.

## Methods
### Metadynamics
Instead of being sampled by basin-hopping[31], which has been widely applied in atmospheric noncovalent interaction clusters[32], the potential energy surfaces of nucleation clusters are sampled by metadynamics (MetaMD)[33,34] due to its remarkable ability to sample high-

energy isomers to prepare the initial data set for further active learning iterations. The bump perturbation can be calculated as[33]

$$V_{\text{bump}}\left(\vec{\mathbf{R}}\right) = \sum_{\alpha} \lambda e^{-\sum_{ij}\left(D_{ij}(\vec{\mathbf{R}})-D_{ij}^{\alpha}\right)^2/(2\sigma^2)} \qquad (1)$$

Here, $\vec{\mathbf{R}}$ represents the atomic coordinates; $\alpha$ sums over snapshots of geometries where the matrix of atomic distances at a given point during the trajectory: $D_{ij} = 1/|\vec{r_{ij}}|$ is the collective variable, where $\vec{r_{ij}}$ are the atomic distances; $i$ and $j$ loop over all the atoms in the frame; $D_{ij}^{\alpha}$ are the previous distance matrices, which we accumulate every $\tau$ femtoseconds (fs); the bumps with bump width $\sigma$ and bump height $\lambda$ are applied to all elements of the contact matrix. The bump width $\sigma$, bump height $\lambda$, and bump time $\tau$ are set to 2.0, 1.0, and 10, respectively, while the MD temperature, time step, and thermostat for the NVT ensemble are set to 600 K, 0.5 fs, and Anderson, respectively. The cluster whose size is within the range of (SA)ₘ(DMA)ₙ (m = 0–4, n = 0–4) is sampled with MetaMD in the Tensormol[35] package interfaced with the PM7 semiempirical method in Gaussian16[36]. To save computational costs, not all cluster sizes within the range are sampled. According to the experimental and theoretical predictions, the sulfuric acid–dimethylamine system tends to grow with a similar number of molecules within the cluster[5]; therefore, for large clusters, those with a difference between the number of acid and base molecules less than or

equal to one are included. For each cluster size, ~50,000 structures are sampled and subsequently selected by the farthest point sampling (FPS) method based on the many-body tensor representation (MBTR) descriptor[37] for further DFT ($\omega$B97XD/6−31++G(d,p)) energy and force labeling. $\omega$B97XD/6−31++G(d,p) is chosen because the systematic benchmark[38] for aerosol nucleation clusters proves its good balance between accuracy and cost. The detailed MetaMD sampling and subsequent DFT calculation procedures are listed in Supplementary Table 2.

### Active learning

Based on the initial data set prepared by MetaMD, an active learning or an on-the-fly strategy[39] is utilized. The MetaMD sampling subset after screening is the initial data set to kick off the active learning iterations. In each iteration, first, 400,000 steps of training with different seeds are conducted to generate four DNN models. Then, the constant-temperature, constant-volume ensemble (NVT) MD simulations are performed in LAMMPS[40] based on trained DNN models. During the MD simulations, four DNN models are utilized to pinpoint the candidate clusters whose error indicators satisfy the threshold range. The error indicator is the maximal standard deviation of the atomic force predicted by the model ensemble. The upper and lower threshold values are 0.50 and 0.35 eV/Å, respectively, indicating that those whose error indicator is below 0.35 eV/Å are regarded as accurate and those whose error indicator is above 0.50 eV/Å are regarded as physically unreasonable. Finally, the energies and forces of the candidate clusters are obtained by $\omega$B97XD/6−31++G(d,p) in the Gaussian16[36] package for training in the next iteration. Notably, the candidate clusters are carved out from the MD trajectory according to the interatomic distance cut-off of 3.5 Å. Here, the clusters are obtained when the shortest interatomic distance between molecules is shorter than the interatomic distance cut-off value to maintain the integrity of the molecular cluster, which is different from the strategy used in a similar work conducted for combustion reactions[15]. The detailed iteration processes are listed in Supplementary Table 3.

### DNN model

The smooth version of the deep potential[41,42] model is conducted in active learning. In deep potential, the potential energy of a molecular cluster is a sum of "atomic energies" $E = \sum_i E_i$, where $E_i$ is determined by the local environment of atom $i$ within a cut-off radius. The model includes two networks: the embedding network and the fitting network. The embedding network is of size (25, 50, 100) and the fitting network is of size (240, 240, 240). The fitting network uses ResNet architecture[43]. The cut-off radius is set to 6.0 Å and the descriptors decay smoothly from 5.8 to 6.0 Å. The learning rate starts at $1.0 \times 10^{-3}$ and exponentially decays every 2000 steps in 400,000 training steps in each active learning iteration. The loss function is defined as a sum of different mean square errors of the DNN predictions for energy and force. The long-range DNN model, Physnet[44], is utilized to train on the final data set for 10,000,000 steps. DeePMD with strictly local descriptors is integrated with LAMMPS, which guarantees high efficiency of MD exploration, so DeePMD is utilized in the active learning iterations. Despite the recent appearance of the long-range version DeePMD (DPLR)[45], we switched to Physnet for production, as preparing maximally localized Wannier centers for DPLR requires a large cell where the molecular electron density decreases to zero on the faces of the cell, which is computationally expensive. Because D3BJ instead of D3 is fitted in Physnet, the final data set is further calculated at the level of $\omega$B97X-D3BJ/6−31++G(d,p) through ORCA 5.0[46] to label structures with the energies and forces as well as the dipole moments. The width of the neural network is controlled by setting the feature space dimensionality and radial basis function number to 128 and 64, respectively, while the neural network depth is controlled by setting the stacked modular building blocks number, residual block number

for atom-wise refinements, residual block number for refinements of proto-message and residual block number in output blocks to 5, 2, 3, and 1, respectively. The cut-off radius for interactions in the neural network is set to 10 Å and long-range interactions are explicitly included by electrostatics and dispersion corrections.

### Molecular dynamics

The collision and evaporation simulations of molecular clusters are conducted under the NVT ensemble at 278 and 300 K through the Atomic Simulation Environment (ASE) with ten SA molecules and ten DMA molecules initially randomly placed in the cubic box with a length of 85 Å. The random positions are given by the packmol[47] package with the stable SA and DMA monomers being the input structures. In each run, MD has performed 100 ps with the COM for each molecule being fixed for equilibration and subsequently 1 ns for production. The cluster positions in the production stage are recorded every 10 fs. The snapshot in Fig. 3a is plotted by VMD[48], while the structure in Fig. 3b is plotted by Chemcraft[49]. The RDF and structural clustering analysis is conducted by freud[50] and ASAP[51], respectively. The proton transfer distance threshold between O and H is set to 1.23 Å. Collison rate constants are derived according to the Poisson distribution feature of the reactive (collision and evaporation) events[16] using the Chem-TraYzer software package[52]. The Poisson-based collision rate constant $k$ can be calculated according to

$$k = \frac{\sum_j N_j}{V \sum_j \left( \sum_i^M C_i \triangle t_i \right)_j} \quad (2)$$

Here, $N_j$ is the collision event number in MD run $j$, $V$ is the MD box volume, $i$ is the subsimulation number in MD run $j$ separated by reactive events, $C$ is the product of reactant concentrations, and $\Delta t$ is the interval between reactive events. The detailed derivation for rate constants and confidence interval of Poisson-based reaction events can be found in the literature[16].

### Cluster kinetics

The molecular cluster kinetics simulations are performed by the home-built Python version[53] of the Atmospheric Cluster Dynamics Code (ACDC)[4] to solve the ordinary differential equations. The collision rate constants are partially replaced by the MD-observed collision event-derived constants, and the remaining collision rate constants are calculated by a hard-sphere collision model. The evaporation rates are calculated assuming a detailed balance based on quantum chemical thermodynamics[11] from the literature[54]. The condensation sink (CS) is set to be $2.6 \times 10^{-3}$ s$^{-1}$ and $2.7 \times 10^{-2}$ s$^{-1}$ to mimic condensation under clean[55] and polluted[56] environments, respectively. In ACDC, the formation rate can be calculated by

$$J = \sum_{i=0}^{4} \sum_{j=0}^{4} \sum_{k=0}^{4} \sum_{l=0}^{4} \beta_{ik,jl} c_{ik} c_{jl} (i+j \geq 4, k+l > 4) \quad (3)$$

Here, $i$ and $j$ refer to the number of SA molecules in each binary collision molecular cluster, and $k$ and $l$ refer to the number of DMA molecules in each binary collision molecular cluster. The time evolution of the cluster concentration $c_i$ can be obtained by solving the birth and death equations given by

$$\frac{dC_i}{dt} = \frac{1}{2} \sum_{j<i} \beta_{j,(i-j)} c_j c_{i-j} + \sum_j \gamma_{(i+j) \to i} c_{(i+j)} - \sum_j \beta_{i,j} c_i c_j - \frac{1}{2} \sum_{j<i} \gamma_{i \to j} c_i + CS \quad (4)$$

Here, $CS$ represents the condensation sink. $\beta_{i,j}$ represents the collision rate constants obtained from the hard-sphere collision model

and is calculated by

$$\beta_{i,j} = \left(\frac{3}{4\pi}\right)^{1/6} \left(\frac{6k_bT}{m_i} + \frac{6k_bT}{m_j}\right)^{1/2} \left(V_i^{1/3} + V_j^{1/3}\right)^2 \tag{5}$$

Here, $T$ represents the temperature, $k_b$ represents the Boltzmann constant, and $m_i$ and $V_i$ represent the mass and volume of cluster $i$, respectively. The evaporation coefficient $\gamma_{(i+j)\to i}$ is calculated by

$$\gamma_{(i+j)\to i,j} = \beta_{i,j}\frac{c_i^e c_j^e}{c_{i+j}^e} = \beta_{i,j}c_{ref}\exp\left(\frac{\triangle G_{i+j} - \triangle G_i - \triangle G_j}{k_bT}\right) \tag{6}$$

Here, $i$ and $j$ are the daughter clusters, $\beta_{i,j}$ is the collision rate constant between $i$ and $j$, $c_i^e$ is the equilibrium concentration of cluster $i$, $\Delta G_i$ is the free energy of formation of cluster $i$ from the constituent monomers, and $c_{ref}$ is the monomer concentration of the reference vapor for which the free energies were calculated.

## Data availability
The training and test data set for DNN, the DNN-FF model, and molecular dynamics trajectories based on DNN-FF are available on figshare (https://doi.org/10.6084/m9.figshare.20968156.v1)[57]. Source data are provided with this paper.

## Code availability
The codes, including metadynamics sampling, active learning, DNN training, molecular dynamics, and cluster kinetics, in addition to the data and scripts to reproduce all the figures in the manuscript and supplementary materials, are available on figshare (https://doi.org/10.6084/m9.figshare.20968156.v1)[57].

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

## Acknowledgements

We thank Linfeng Zhang, Jinzhe Zeng, and other deep potential community contributors for the diligent support of DeePMD and DP-GEN. We thank Linfeng Zhang for the very helpful comments on the manuscript. We thank John E. Herr for the help with the metadynamics test and analysis. We thank Roope Halonen and Bernhard Reischl for sharing with us the input files to reproduce sulfuric acid monomer collision simulations. We acknowledge the support of the GPU cluster built by the MCC Lab of Information Science and Technology Institution, USTC. This work was supported by the National Natural Science Foundation of China (Grant No. 41877305).

## Author contributions

S.J. proposed the workflow and conducted metadynamics, active learning, DNN training, as well as molecular dynamics and analysis. Y.-R.L., T.H., Y.-J.F., C.-Y.W., and Z.-Q.W. helped write and edit the paper. B.-J.G., Q.-S.L., and W.-R.G. improved the terminology expressions of the manuscript. W.H. helped with the design of the workflow. All authors commented on the manuscript.

## Competing interests

The authors declare no competing interests.
