## [Peer Review File · Nature Communications]

Towards fully ab initio simulation of atmospheric aerosol nucleationREVIEWER COMMENTS

Reviewer #1 (Remarks to the Author):

This paper describes a workflow for developing machine learning potentials to study aerosol nucleation. Specifically, the authors start by training a potential on configurations sampled from semi-empirical force fields using the metadynamics enhanced sampling method, followed by an active learning strategy in which the potential is used to generate new configurations to be included in the training set. This procedure results in a diverse set of configurations consisting of different cluster compositions. The potential is then used to perform molecular dynamics simulations and extract information about aerosol nucleation kinetics, going beyond the hard-sphere models previously used in the literature, as well as gaining insights into the structure of the clusters. I think such an approach is of interest not only to those studying aerosol nucleation but also to the broader molecular simulation community.

The manuscript is well organized into sections related to each step of the workflow. However, the manuscript is poorly written, and as a result, it is often difficult to understand the details of the messages the authors want to convey. I would highly recommend having it proofread by a native English speaker or by specialized services. If not this, I suggest at least using some grammar and syntax checking software, as it currently contains several typos and grammatical errors. A few of them are reported below:

line 6: are → is

lines 48,87,155: is → are

35: though → through

68: interpolate → interpolative?

71: application value → applicability?

112: pronated → protonated ?

181: complication → complicated ?

409: uncertain → uncertainty ?

Beyond this, throughout the manuscript, the authors make bold claims about the universality and robustness of their workflow, which, they believe, is capable of moving aerosol simulations toward fully ab initio accuracy. This would represent a significant advance in molecular simulations of aerosol, however for the authors to support these claims, they must provide additional evidence, especially regarding the issues detailed below.

- Regarding the DNN force field benchmark: the fact that the simulations do not explode or that the molecules do not break into atoms is a necessary but not sufficient condition to prove that a potential is accurate. The authors note that the DNN-FF potential is more accurate than empirical potentials, as would be expected from a DNN potential trained on a single system. However, the dimer detachment curves (Fig. 2a-c) show a not very good agreement between DFT and DNN-FF. While the smoothness could be related to the truncation of the long-range electrostatics, its discrepancies within the NN cutoff are surprising. This might indicate a problem in the way the training set structures are collected. If I understand correctly, the configurations used to train the potential in the passive learning step were all sampled with an empirical potential (PM7). Although the energies are then recalculated at the DFT level, this does not guarantee that all the structures relevant to DFT are collected correctly. I would suggest checking this point, for example by running short ab-initio MD (AIMD) simulations and comparing structural properties with the configurations sampled with PM7. If there is not enough overlap in the sampled structures, adding configurations from AIMD might indeed be necessary to reproduce correctly the detachment curves.

- The authors argue that one of the main challenges of aerosol molecular simulations is the non-reactivity of standard interatomic potentials. However, they provide no evidence that the presented approach can provide an accurate description of the high-energy regions. In a recent paper on the construction of machine learning potentials for reactive events [Yang, Parrinello et. al., Catalysis Today, 2021], it was reported that appropriate sampling of the transition states region is critical to obtain a potential that can provide accurate reaction rates, and that active learning as also

implemented in the present work is not always sufficient. I would suggest confirming the accuracy of the high-energy isomers predicted by DNN-FF [lines 121-122] with DFT calculations. In addition, it would be appropriate to compare the description for DFT and DNN-FF of the transition states related to collision events, for example through nudged elastic band calculations. This would strengthen the argument that the potential is capable of accurately describing reactive events.

- In both the benchmark section (lines 88-89) and the aerosol kinetics one (lines 142-144), the authors note that the Deep Potential model they are using to represent potential energy truncates long-range electrostatics at a short cutoff distance. The authors say that "this shortcoming doesn't affect the validity and application value of the framework". However, this statement is not substantiated. Instead, it seems to be an important issue for this type of system, as seen for example in the dimer detachment curves and as noted by the authors regarding the collision rates. While this limitation does not prevent qualitative information from being extracted, I think the authors need to discuss this limitation in more detail instead of saying, "not a problem". Also, this makes the choice of the cutoff parameter for the potential fitting more delicate. Therefore, it might be worth checking if and how the results are affected by changing this parameter.

Minor comments:

[Line 9] "Here we proposes a system transferable framework to...": in ML jargon, transferability means that a model trained on one system can be applied on a different one. Since this is not the case, I would suggest replacing it with "a general workflow that is also applicable to other systems."

[36] The term "passive learning" is uncommon, and can only be found in a few references on machine learning potentials. Please describe it to improve clarity.

[68-70] "Due to the interpolate nature of DNN, size scalability is still limited for extending only several molecules in this case. However, considering the rare event nature of aerosol nucleation, direct MD for ambient or laboratory aerosol nucleation with high concentration is very expensive challenging the application value of DNN trained model."

These sentences are not clear, could you please clarify what you mean?

[76] Figure 2. Does panel b include both the configurations used for training and testing? If yes, consider highlighting them differently. Also regarding panels e-f you need to describe on which cluster compositions they were calculated.

[92] There is a possible typo: the RMSE value according to Fig. 2d should be 7×10^{-3} and not 7×10^{-2} .

[94-96] The statement "The size extension to (SA)10(DMA)10 gives quite good accuracy with $2.98 \cdot 10^{-1} \text{ eV/\AA}$ for all cluster sizes considered here(Fig.2f) and $3.15 \cdot 10^{-1} \text{ eV/\AA}$ for (SA)10(DMA)10alone (Fig.2g)" is somewhat convoluted. Please consider rewriting it differently.

[130] Please indicate what CS stands for since it is only written in the methods section.

[189-193] The reference cited (38) is not to the original metadynamics [Laio, Parrinello, PNAS 2002], rather to a paper that used that method to sample the PES. I think it might be appropriate to cite both of them. Also, note that in the way it is used by Ref. 38 there is not a single collective variable as reported, rather the "bumps" are applied to all elements of the contact matrix.

[211-214] If I understand correctly, the authors use a two-threshold cutoff, where only configurations for which the prediction uncertainty is within the two thresholds are added to the training set. Instead, the sentence "The threshold ranges from 0.35 eV/A to 0.5 eV/A ..." seems to imply that the thresholds have been changed within the range [0.35-0.5].

Reviewer #2 (Remarks to the Author):

The manuscript presents an interesting deep learning-based framework to investigate atmospheric aerosol nucleation by means of molecular dynamics simulations. Here, a reactive deep neural network force field (DNN-FF) is obtained by constructing an initial data set of representative configurations using a DFT (PM7) MD simulation coupled with metadynamics and by further enriching this set with other structures coming from a so-called “active learning” algorithm. Then, the FF is obtained by training a deep neural network potential on the set of configurations previously generated. From the trajectories thus obtained, the authors got relevant information on the rate constants and acid-base behavior of these clusters.

As it is also reported in the first section, aerosol nucleation is a process at the heart of climate studies and its understanding is a fundamental step in the comprehension of such phenomena. However, the paper in its current form is not always easy to follow and I think a major revision is needed to make it publishable.

My main concerns are related to the many computational details that are missing and that make reproducibility of this work nearly impossible. Furthermore, a rigorous validation of the FF is missing making any outcome derived from such simulations unpredictable. In particular, the construction of the data set and the following sampling obtained from the DNN-based FF present some critical issues that need further clarification.

Passive Learning:

- About metadynamics, since this method relies on the identification of a lower-dimensional description of the system, the authors should clarify which collective variables they used. In Eq. 1 many variables and indices are not introduced. The external bias potential seems to be a function of R that has been never mentioned. What does R stand for? Also the distance matrix D_{ij} has never been introduced. On which sites (i and j) does the summation run?
- What is the deposition rate of the gaussian hills?
- Since Gaussian16 cannot perform MetaD simulations, the authors should indicate if a development version of Gaussian or an external plugin was used to run such simulations.

DNN FF based MD:

- Each system has been tested at 278 and 300 K. Is there a reason for having chosen these two values? The authors said that no differences can be observed between these two temperatures (line 145). It could be interesting to test if this universal behavior is also observed for larger temperature intervals.
- Another aspect that needs to be validated and reported, is the accuracy of the interatomic potential obtained. The authors said that “unphysical behaviors like molecules dissociating into atoms” were not observed (line 91). I think that this is not enough to prove the validity of the FF and further comparisons with higher levels of theory MD simulations should be done.
- During the MD simulations, a proton transfer event is observed at around 963 ps (line 123 and Extended Data Figures 4 and 5). A single reactive event indicates that the simulation is far from ergodicity and that it must be extended by the time necessary to make the relative population of each thermodynamic state independent of the trajectory time length. Has this test been done for each MD simulation? Since the cost of a DNN-FF MD simulation should be the same as a classical one, why did not the authors extend the simulations for more than 1 ns?

Minor points:

- In Figure 2C, the binding energy is plotted only from 1.0 up to 1.7 Å. On this scale the equilibrium distance is not even observable and the energies are negative over the entire range. I suggest showing a larger range as done for Figures 2A and 2B.
- In Figure 3D, axis labels and tick marks are absent.
- (line 57) “tests show purely based on passive learning provided data set cannot provide a robust DNN model as unphysical behaviors are observed”. I suggest adding a citation or providing such evidence.

Reviewer #3 (Remarks to the Author):

In "A universal deep learning-based framework towards fully ab initio simulation of atmospheric aerosol nucleation" the authors describe a framework for constructing neural network-based force fields for simulating atmospheric aerosol nucleation.

Unfortunately, I fail to see the novelty of the presented approach. The described framework consists of sampling an initial dataset using metadynamics, augmenting the initial data by running MD simulations with an ensemble of models (query-by-committee), and then training a DeePMD model on the full dataset. All of these methods are well-established individually, and even the combination of these (or similar) steps is standard procedure. In fact, a general approach for constructing machine learned force fields, which is almost identical to the proposed framework, is already described in a recent review article (which the authors even cite, see Ref. 27).

In addition to the lack of novelty, the presented results are not very convincing in my opinion. For example, the DNN-predicted dimer binding curves in Fig. 2a and 2b show strong oscillatory artefacts in the long-range region. Additionally, the correlation plots in 2e, 2f, and 2g show large outliers (several eV/Å) and for many of the clusters, the energy RMSEs presented in 2d are on the order of several kcal/mol. Further, there are some questionable statements. For example, the authors claim that the lack of electrostatic interactions beyond the cutoff does not affect the validity of the proposed framework (line 88 and line 142), even though they admit that the CEF is likely underestimated due to the lack of long-range interactions (line 143). Given that this seems to be crucial, it is unclear to me why the authors did not augment their DNN with long-range electrostatics, in particular because several existing DNNs already do this routinely (see e.g. 10.1021/acs.jctc.9b00181). I would expect that this would greatly improve their results.

There are also some issues with the structure of the text. To give an example, in the Results section, the authors jump straight to a discussion of passive learning, without any general overview of the individual components of their framework (apart from Fig.1). Even though the methods section provides further details, I think a small paragraph with a general overview would make this section much more readable.

Finally, there are many language issues (colloquialism/missing words/wrong grammar), which make the manuscript hard to read. For example (this list is far from exhaustive):

- line 9: "we proposes"
- line 19: "Emergence in 2011 and then broadly employed ..."
- line 25: "However, collision rate constants, derived from simple hard sphere model, are still very rough, which accuracy is far from ..."
- line 44: "... to prepare data set ..."
- line 58: "... that's why ..."
- line 68: "Due to the interpolate nature of DNN ..."
- line 117: "The more inner position ..."
- line 124: "... tons of proton transfer events ..."
- line 155: "... ranges from one to six or so."

I advise additional proofreading, preferably by a native speaker.

In summary, I recommend that the authors fix the major flaws and try to submit the paper to a different journal. Even without any of the above-mentioned issues, I don't think this manuscript is suitable for publication in Nature Communications due to the lack of novelty.

Reviewer #1 (Remarks to the Author):

This paper describes a workflow for developing machine learning potentials to study aerosol nucleation. Specifically, the authors start by training a potential on configurations sampled from semi-empirical force fields using the metadynamics enhanced sampling method, followed by an active learning strategy in which the potential is used to generate new configurations to be included in the training set. This procedure results in a diverse set of configurations consisting of different cluster compositions. The potential is then used to perform molecular dynamics simulations and extract information about aerosol nucleation kinetics, going beyond the hard-sphere models previously used in the literature, as well as gaining insights into the structure of the clusters. I think such an approach is of interest not only to those studying aerosol nucleation but also to the broader molecular simulation community.

Reply: We thank the reviewer for the very positive assessment of our work.

The manuscript is well organized into sections related to each step of the workflow. However, the manuscript is poorly written, and as a result, it is often difficult to understand the details of the messages the authors want to convey. I would highly recommend having it proofread by a native English speaker or by specialized services. If not this, I suggest at least using some grammar and syntax checking software, as it currently contains several typos and grammatical errors. A few of them are reported below:

line 6: are → is

lines 48,87,155: is → are

35: though → through

68: interpolate → interpolative?

71: application value → applicability?

112: pronated → protonated ?

181: complication → complicated ?

409: uncertain → uncertainty ?

Reply: We thank the reviewer for the comments, and we are very sorry for these errors. The grammar/syntax/typos errors have been corrected as suggested, and the manuscript has been further polished by author services from Springer Nature.

Beyond this, throughout the manuscript, the authors make bold claims about the universality and robustness of their workflow, which, they believe, is capable of moving aerosol simulations toward fully ab initio accuracy. This would represent a significant advance in molecular simulations of aerosol, however for the authors to support these claims, they must provide additional evidence, especially regarding the issues detailed below.

- Regarding the DNN force field benchmark: the fact that the simulations do not explode or that the molecules do not break into atoms is a necessary but not sufficient condition to prove that a potential is accurate. The authors note that the DNN-FF potential is more accurate than empirical potentials,

as would be expected from a DNN potential trained on a single system. However, the dimer detachment curves (Fig. 2a-c) show a not very good agreement between DFT and DNN-FF. While the smoothness could be related to the truncation of the long-range electrostatics, its discrepancies within the NN cutoff are surprising. This might indicate a problem in the way the training set structures are collected. If I understand correctly, the configurations used to train the potential in the passive learning step were all sampled with an empirical potential (PM7). Although the energies are then recalculated at the DFT level, this does not guarantee that all the structures relevant to DFT are collected correctly. I would suggest checking this point, for example by running short ab-initio MD (AIMD) simulations and comparing structural properties with the configurations sampled with PM7. If there is not enough overlap in the sampled structures, adding configurations from AIMD might indeed be necessary to reproduce correctly the detachment curves.

Reply: We thank the reviewer for the very helpful hints. We rechecked the dimer detachment curves and found that the previously used characterization parameter, “binding energy”, was wrongly used. Using binding energy, both the dimers and monomers should be optimized under the same level of method, but we did not optimize the monomers. Then, we looked up the literature¹⁻³ about using machine learning methods to build force fields for molecular clusters and found that typically, “relative energy” instead of “binding energy” was used. Here, relative energy represents the isomer energy minus the energy of the most stable isomer. Therefore we trained a deep neural network (DNN) model using Physnet⁴ with the same data set we used for DPMD and benchmarked the model performance by relative energy. Notably, to properly consider long-range interactions, we switched DPMD to Physnet. The details about the Physnet model parameters have been added to the Methods section.

Fig. R1. Dimer detachment curves calculated by DFT and deep neural network (DNN) models trained by Physnet based on a data set prepared with metadynamics and an active learning strategy. DNN, DNN_elec, DNN_disp and DNN_short represent the model with electrostatics and dispersions, the model with electrostatics and without dispersion, the model without electrostatics and with dispersion, and the model without electrostatics and dispersion, respectively.

We plot the dimer detachment curves with relative energy and find that the large energy gap within the cut-off range is significantly decreased (Fig. R1). After including long-range corrections, the dimer relative energy prediction error also decreases significantly, for instance, from 4.06 kcal/mol to 0.96 kcal/mol for (SA)₁(DMA)₁. In addition, the smoothness of the curves also improves. Therefore, after adding long-range interactions, the DFT dimer detachment curves can be accurately reproduced by DNN-based force fields.

Despite the encouraging performance of the long-range DNN model based on the previously used data set, we still recheck the structures sampled by PM7 as suggested. To check whether the structures given by PM7-driven metadynamics (PM7 structures in short) are problematic, we compare the energetics from the PM7 structures with those from DFT-driven metadynamics (DFT structures). The DFT and PM7 structures are both obtained through metadynamics/molecular dynamics by keeping the other parameters (initial structure, initial velocity, temperature, thermostat, etc.) the same.

Fig. R2. Energetics comparison between the PM7 and DFT structures. Here metadynamics (MetaMD) is conducted with a bump height and width of 1.0 and 2.0, respectively. Both molecular dynamics (MD) and MetaMD are run under the NVT ensemble with the Andersen thermostat at $T=600$ K. The root mean squared errors (RMSEs) are calculated by comparing the others with DFT_structures_MD. PM7_structures_MetaMD, PM7_structures_MD, DFT_structures_MetaMD and DFT_structures_MD represent PM7-driven MetaMD sampling structures, PM7-driven MD

sampling structures, DFT-driven MetaMD sampling structures and DFT-driven MD sampled structures, respectively. Energies on PM7_structures_MetaMD and PM7_structures_MD are both further calculated by DFT.

From Fig.R2, we can see that the root mean squared error (RMSE) of DFT energies on the PM7 structures are systematically higher by several kcal/mol than the RMSE of DFT energies on the DFT structures. Removing bumps by switching MetaMD to MD reduces the RMSE by approximately one kcal/mol, so the high RMSE (6.44 kcal/mol) of PM7_structures_MetaMD is mainly contributed by PM7 instead of MetaMD bumps. Therefore, we speculate that the low energy region is probably not sufficiently sampled.

Therefore, we further sample the three dimer systems for 5 ps with a time step of 1 fs at 300 K by *ab initio* molecular dynamics (AIMD). We extract one structure every 10 fs, so we obtain 1500 AIMD frames in total, to be added into the original data set for training.

Fig. R3. Dimer detachment curves calculated by DFT and deep neural network (DNN) models trained by Physnet based on dataset prepared with metadynamics and an active learning strategy in addition to the AIMD trajectory consisting of 1500 frame dimers. DNN, DNN_elec, DNN_disp and DNN_short represent the model with electrostatics and dispersions, the model with electrostatics and without dispersion, the model without electrostatics and with dispersion, and the model without electrostatics and dispersion, respectively.

As seen in Fig. R3, after adding the AIMD structures, for dimer detachment curves, the DNN model gives not only a lower RMSE but also lower max error values if we compare the results with those in Fig. R1. For instance, for (SA)₁(DMA)₁, under the long-range model, the energy RMSE decreases from 0.96 kcal/mol to 0.49 kcal/mol, and the energy max error also decreases from 2.10 kcal/mol to 0.82 kcal/mol.

Therefore, we update the original data set by adding those AIMD structures and train the DNN model with Physnet.

- The authors argue that one of the main challenges of aerosol molecular simulations is the non-reactivity of standard interatomic potentials. However, they provide no evidence that the presented approach can provide an accurate description of the high-energy regions. In a recent paper on the construction of machine learning potentials for reactive events [Yang, Parrinello et. al., Catalysis Today, 2021], it was reported that appropriate sampling of the transition states region is critical to obtain a potential that can provide accurate reaction rates, and that active learning as also implemented in the present work is not always sufficient. I would suggest confirming the accuracy of the high-energy isomers predicted by DNN-FF [lines 121-122] with DFT calculations. In addition, it would be appropriate to compare the description for DFT and DNN-FF of the transition states related to collision events, for example through nudged elastic band calculations. This would strengthen the argument that the potential is capable of accurately describing reactive events.

Reply: We thank the reviewer for the important suggestion. Multiple trials are conducted to find transition states (TS) by nudged elastic band (NEB) calculations for the collision between sulfuric acid (SA) monomer and dimethylamine (DMA) monomer; however, all trials indicate a barrier-free process without TS. Nevertheless, we could pull out the NEB TS trajectory and compare the energetics for DFT and DNN, as shown below.

Fig. R4. Relative energy curves on the NEB TS trajectory under the level of DFT and DNN. Three representative structures show the process of proton transfer and hydrogen bond formation.

Proton transfer and hydrogen bond formation events are captured. During the whole process, the relative energy prediction errors between DNN and DFT are quite uniform, with RMSE and max errors of 0.18 kcal/mol and 0.58 kcal/mol, respectively.

Furthermore, we conduct AIMD simulations for the collision of the SA monomer and DMA monomer to check the DNN model accuracy. AIMD simulations are performed under the NVT ensemble at $T=300$ K for 5 ps with a time step of 1 fs, producing 5000 frame structures, whose relative energy curves are shown below. The SA monomer and DMA monomer are both placed with a distance of 8 Å initially before AIMD.

Fig. R5. Relative energy (RE) and ΔRE for the AIMD trajectory calculated by DFT, DNN with electrostatics and dispersion correction and PM7. ΔRE means DNN/PM7 relative energy subtracted from DFT relative energy for the same structure from the AIMD trajectory.

As shown above, the DNN model can more closely follow the relative energy curve shape of DFT than PM7. In addition, the RMSE and max error of the DNN model are much smaller than those for

PM7, which is also reflected in ΔRE .

However, since the strategies used in both works are similar, we believe that the methodology proposed in [Yang, Parrinello et. al., *Catalysis Today*, 2021] is probably vital to deal with nucleation systems with apparent barriers, such as the sulfuric acid-ammonia system. Therefore, we added this information to the manuscript: *The workflow we propose here works for strong acid and strong base nucleation systems, which are the most significant systems in the aerosol nucleation field due to their strong nucleation ability. However, for systems with apparent barriers, such as the sulfuric acid-ammonia system, it is probably vital to utilize the strategy⁵ to introduce transition state configurations to obtain a uniformly accurate model.*

- *In both the benchmark section (lines 88-89) and the aerosol kinetics one (lines 142-144), the authors note that the Deep Potential model they are using to represent potential energy truncates long-range electrostatics at a short cutoff distance. The authors say that "this shortcoming doesn't affect the validity and application value of the framework". However, this statement is not substantiated. Instead, it seems to be an important issue for this type of system, as seen for example in the dimer detachment curves and as noted by the authors regarding the collision rates. While this limitation does not prevent qualitative information from being extracted, I think the authors need to discuss this limitation in more detail instead of saying, "not a problem". Also, this makes the choice of the cutoff parameter for the potential fitting more delicate. Therefore, it might be worth checking if and how the results are affected by changing this parameter.*

Reply: We thank the reviewer for the suggestion about the long-range issue. We agree with the reviewer that long-range interactions should be properly considered, so after obtaining the data set through metadynamics coupled with an active learning strategy in addition to the AIMD trajectory, we switch the short-range DPMD model to the Physnet⁴ model. We do notice that very recently, the long-range version of DPMD² was proposed; however, preparing maximally localized Wannier centers (MLWCs) for electrostatic energy requires calculations conducted in the first principles software such as VASP. Calculating gas phase molecular clusters under the first principles software requires a cell with a large enough size, so the computational costs are large and improper setting of cell size could also introduce artifacts. Therefore, despite the great potential of the long-range DPMD, we still adopt Physnet for including the long-range interactions.

Here are the changes to the Methods section in the manuscript: *The long-range DNN model, Physnet⁴, is utilized to train on the final data set for 10,000,000 steps. Because D3BJ instead of D3 is fitted in Physnet, the final data set is further calculated at the level of $\omega B97X-D3BJ/6-31++G(d,p)$ through ORCA 5.0⁶ to label structures with the energies and forces as well as the dipole moments. The width of the neural network is controlled by setting the feature space dimensionality and radial basis function number to 128 and 64, respectively, while the neural network depth is controlled by setting the stacked modular building block number, residual block number for atomwise refinements, residual block number for refinements of proto-message and residual block number in output blocks to 5, 2, 3 and 1, respectively. The cut-off radius for interactions in the neural network is set to 10 Å, and long-range interactions are explicitly included by electrostatics and dispersion corrections.*

Minor comments:

[Line 9] "Here we proposes a system transferable framework to...": in ML jargon, transferability means that a model trained on one system can be applied on a different one. Since this is not the case, I would suggest replacing it with "a general workflow that is also applicable to other systems."

Reply: We thank the reviewer for the suggestion. We recognize the review's opinion that "workflow" is more suitable than "framework", so all the places mentioned "framework" in the manuscript have been replaced with "workflow". In addition, the title "*A universal deep learning-based framework towards fully ab initio simulation of atmospheric aerosol nucleation*" has been changed to "*Towards fully ab initio simulation of atmospheric aerosol nucleation*" to make it more concise.

[36] The term "passive learning" is uncommon, and can only be found in a few references on machine learning potentials. Please describe it to improve clarity.

Reply: We thank the reviewer for the suggestion. We abandon "passive learning" and replace it with "metadynamics" for clarity.

[68-70] "Due to the interpolate nature of DNN, size scalability is still limited for extending only several molecules in this case. However, considering the rare event nature of aerosol nucleation, direct MD for ambient or laboratory aerosol nucleation with high concentration is very expensive challenging the application value of DNN trained model."

These sentences are not clear, could you please clarify what you mean?

Reply: We thank the reviewer for the comment. To make it clear, those sentences are replaced with "Due to the interpolative nature of DNNs, high accuracy could be maintained for clusters up to (SA)₁₀(DMA)₁₀. The accuracy for clusters beyond (SA)₁₀(DMA)₁₀ is unknown, but we expect a further decrease in accuracy. The ultimate goal of simulating atmospheric aerosol nucleation is to conduct MD under ambient or laboratory conditions, where the cluster size can easily go beyond the interpolation regime, so how to apply the DNN model with size limited accuracy to atmospheric nucleation becomes a problem that needs to be addressed."

[76] Figure 2. Does panel b include both the configurations used for training and testing? If yes, consider highlighting them differently. Also regarding panels e-f you need to describe on which cluster compositions they were calculated.

[92] There is a possible typo: the RMSE value according to Fig. 2d should be 7×10^{-3} and not 7×10^{-2} .

[94-96] The statement "The size extension to (SA)₁₀(DMA)₁₀ gives quite good accuracy with 2.98.10-1eV/Å for all cluster sizes considered here(Fig.2f) and 3.15. 10-1eV/Å for (SA)₁₀(DMA)₁₀alone (Fig.2g)" is somewhat convoluted. Please consider rewriting it differently.

Reply: We thank the reviewer for the comments about Figure 2. We believe that the reviewer refers to panel d instead of panel b for the question of the configurations used for training and testing. In panel d, all the configurations are used for testing. To make it clear, we added this information to

the caption of Fig. 2: *d*, Energy and force root mean squared error (RMSE) values in the interpolation and extrapolation regimes of the test set. In addition, here we draw Figure 2 by combining the model's energy and force RMSE performance in one plot, as shown in the manuscript, to make the figure more concise. The typo and statement have been changed accordingly as suggested.

[130] Please indicate what CS stands for since it is only written in the methods section.

Reply: We thank the reviewer for the suggestion. The explanation for CS is added in Fig. 4 caption as suggested: ... where T and CS are for temperature and condensation sink coefficient, respectively.

[189-193] The reference cited (38) is not to the original metadynamics [Laio, Parrinello, PNAS 2002], rather to a paper that used that method to sample the PES. I think it might be appropriate to cite both of them. Also, note that in the way it is used by Ref. 38 there is not a single collective variable as reported, rather the "bumps" are applied to all elements of the contact matrix.

Reply: We thank the reviewer for the suggestion. We agree with the reviewer that the original metadynamics should be added and has been added as suggested. The description of metadynamics is changed to "... the bumps with bump width σ and bump height λ are applied to all elements of the contact matrix".

[211-214] If I understand correctly, the authors use a two-threshold cutoff, where only configurations for which the prediction uncertainty is within the two thresholds are added to the training set. Instead, the sentence "The threshold ranges from 0.35 eV/Å to 0.5 eV/Å ..." seems to imply that the thresholds have been changed within the range [0.35-0.5].

Reply: We thank the reviewer for the comment. It is a two-threshold cut-off, so the sentence has been changed to "The upper and lower threshold values are 0.50 eV/Å and 0.35 eV/Å, respectively ...".

Reviewer #2 (Remarks to the Author):

The manuscript presents an interesting deep learning-based framework to investigate atmospheric aerosol nucleation by means of molecular dynamics simulations. Here, a reactive deep neural network force field (DNN-FF) is obtained by constructing an initial data set of representative configurations using a DFT (PM7) MD simulation coupled with metadynamics and by further enriching this set with other structures coming from a so-called "active learning" algorithm. Then, the FF is obtained by training a deep neural network potential on the set of configurations previously generated. From the trajectories thus obtained, the authors got relevant information on the rate constants and acid-base behavior of these clusters.

As it is also reported in the first section, aerosol nucleation is a process at the heart of climate studies and its understanding is a fundamental step in the comprehension of such phenomena. However, the paper in its current form is not always easy to follow and I think a major revision is needed to make it publishable.

Reply: We thank the reviewer for the recommendation. The manuscript was rewritten by us and then further edited for proper English language, grammar, punctuation, spelling, and overall style by Springer Nature author services.

My main concerns are related to the many computational details that are missing and that make reproducibility of this work nearly impossible. Furthermore, a rigorous validation of the FF is missing making any outcome derived from such simulations unpredictable. In particular, the construction of the data set and the following sampling obtained from the DNN-based FF present some critical issues that need further clarification.

Reply: We thank the reviewer for the comments. We added the computational details as suggested below and uploaded all the data and codes to figshare (<https://figshare.com/s/d8a6b17457f738156cf0>) for reproducibility. We will answer the reviewer question about validation and clarification of DNN-FF as follows.

Passive Learning:

- About metadynamics, since this method relies on the identification of a lower-dimensional description of the system, the authors should clarify which collective variables they used. In Eq. 1 many variables and indices are not introduced. The external bias potential seems to be a function of R that has been never mentioned. What does R stand for? Also the distance matrix D_{ij} has never been introduced. On which sites (i and j) does the summation run?

Reply: We thank the reviewer for the comments and we are very sorry for the confusion caused by missing the important technical details. Therefore, the descriptions about metadynamics are changed to the following: Here, \vec{R} represents the atomic coordinates; α sums over snapshots of geometries where the matrix of atomic distances at a given point during the trajectory: $D_{ij} = 1/|\vec{r}_{ij}|$ is the collective variable, where \vec{r}_{ij} are the atomic distances; i and j loop over all the atoms in the frame; D_{ij}^α are the previous distance matrices, which we accumulate every τ femtoseconds (fs); the bumps with bump width σ and bump height λ are applied to all elements of the contact matrix. The bump width σ , bump height λ and bump time τ are set to 2.0, 1.0 and 10, respectively, while the MD temperature, time step and thermostat for the NVT ensemble are set to 600 K, 0.5 fs and Anderson, respectively. In addition, all the metadynamics input scripts are uploaded to figshare (<https://figshare.com/s/d8a6b17457f738156cf0>).

- What is the deposition rate of the gaussian hills?

Reply: We thank the reviewer for the comment. The deposition rate of the Gaussian hills is $1/\tau$, where we accumulate the previous distance matrices every τ femtoseconds (fs), as we explained above in the Methods section.

- Since Gaussian16 cannot perform MetaD simulations, the authors should indicate if a development version of Gaussian or an external plugin was used to run such simulations.

Reply: We thank the reviewer for the suggestion. MetaMD simulations are conducted by the MetaMD module in the open-source TensorMol package; therefore, we add this in the Methods section: *The cluster whose size is within the range of $(SA)_m(DMA)_n$ ($m=0-4$, $n=0-4$) is sampled with MetaMD in the Tensormol³ package interfaced with the PM7 semiempirical method in Gaussian16⁷.*

DNN FF based MD:

- *Each system has been tested at 278 and 300 K. Is there a reason for having chosen these two values? The authors said that no differences can be observed between these two temperatures (line 145). It could be interesting to test if this universal behavior is also observed for larger temperature intervals.*

Reply: We thank the reviewer for the suggestion. We added the reason in Fig. 4 caption: *A temperature of 300 K is chosen to resemble the conditions of standard temperature and pressure (STP), while 278 K represents typical spring-time conditions at the boreal forest site in Hyttiälä⁸, the flagship observatory in the new particle formation field and winter-time conditions for observation in Beijing⁹.*

- *Another aspect that needs to be validated and reported, is the accuracy of the interatomic potential obtained. The authors said that “unphysical behaviors like molecules dissociating into atoms” were not observed (line 91). I think that this is not enough to prove the validity of the FF and further comparisons with higher levels of theory MD simulations should be done.*

Reply: We thank the reviewer for the comment. We agree with the reviewer that it is not enough to validate the DNN-FF that unphysical behaviors such as molecules dissociating into atoms are not observed. Further comparisons with higher levels of theory MD simulations should be performed; therefore, we conducted AIMD simulations for the collision of the SA monomer and DMA monomer to check the DNN model accuracy. The figure and discussions are given to reply to Reviewer #1, in case the reviewer cannot see the results, we add here as follows.

AIMD simulations are performed under the NVT ensemble at T=300 K for 5 ps with a time step of 1 fs, producing 5000 frame structures, whose relative energy curves are shown below. The SA monomer and DMA monomer are both placed with a distance of 8 Å initially before AIMD.

Fig. R5. Relative energy (RE) and ΔRE for the AIMD trajectory calculated by DFT, DNN with electrostatics and dispersion correction and PM7. ΔRE means DNN/PM7 relative energy subtracted from DFT relative energy for the same structure from the AIMD trajectory.

As shown above, DNN-FF can more closely follow the relative energy curve shape of DFT than PM7. In addition, the RMSE and max error of DNN-FF are much smaller than those for PM7, which

is also reflected in Δ RE. Notably, the RMSE of DNN-FF is only 0.41 kcal/mol, far less than 1 kcal/mol, so-called chemical accuracy, proving the high accuracy of DNN-FF.

- *During the MD simulations, a proton transfer event is observed at around 963 ps (line 123 and Extended Data Figures 4 and 5). A single reactive event indicates that the simulation is far from ergodicity and that it must be extended by the time necessary to make the relative population of each thermodynamic state independent of the trajectory time length. Has this test been done for each MD simulation? Since the cost of a DNN-FF MD simulation should be the same as a classical one, why did not the authors extend the simulations for more than 1 ns?*

Reply: We thank the reviewer for the comments. Notably, Extended Data Figures 4 and 5 are now included in the Supplementary Materials as Supplementary Figures 2 and 3. A single reactive event is observed before because we only plot the processes from the formation of (SA)₆(DMA)₆ to the end of 1 ns and most of the reactive events occur before the collision towards forming (SA)₆(DMA)₆. In addition, the motivation is not to capture as many reactive events as possible but to observe whether the collision would affect the hydrogen bonds and transferred proton previously formed. We find that at 1 ns, mostly one large cluster is formed, making the longer simulation unnecessary.

Minor points:

- *In Figure 2C, the binding energy is plotted only from 1.0 up to 1.7 Å. On this scale the equilibrium distance is not even observable and the energies are negative over the entire range. I suggest showing a larger range as done for Figures 2A and 2B.*

Reply: We thank the reviewer for the suggestion. We agree with the reviewer about that and revise the figure as suggested.

- *In Figure 3D, axis labels and tick marks are absent.*

Reply: We thank the reviewer for the comment. In Fig. 3d, the structural similarity of isomers is given by kernel principal component analysis (KPCA) using a global Smooth Overlap of Atomic Positions (SOAP) kernel. In KPCA, the first few eigenvectors of the design matrix, which form the axes of the plot, are also called “principal components”, PCs. These eigenvectors are not physically meaningful like hand-craft collective variables, so typically the axis and ticks are hidden for making the figure concise, which could be seen in the work¹⁰ where the methodology was originally proposed.

- *(line 57) “tests show purely based on passive learning provided data set cannot provide a robust DNN model as unphysical behaviors are observed”. I suggest adding a citation or providing such evidence.*

Reply: We thank the reviewer for the comment. We think it is improper to say that “purely based on passive learning provided data set cannot provide a robust DNN model as unphysical behaviors are observed” even though we did find that in our cases, without the aid of active learning, we didn’t get a robust model. If more metadynamics samplings are performed, it is possible to obtain an

accurate model. Therefore, we deleted the sentence in the manuscript.

Reviewer #3 (Remarks to the Author):

In "A universal deep learning-based framework towards fully ab initio simulation of atmospheric aerosol nucleation" the authors describe a framework for constructing neural network-based force fields for simulating atmospheric aerosol nucleation.

Unfortunately, I fail to see the novelty of the presented approach. The described framework consists of sampling an initial dataset using metadynamics, augmenting the initial data by running MD simulations with an ensemble of models (query-by-committee), and then training a DeePMD model on the full dataset. All of these methods are well-established individually, and even the combination of these (or similar) steps is standard procedure. In fact, a general approach for constructing machine learned force fields, which is almost identical to the proposed framework, is already described in a recent review article (which the authors even cite, see Ref. 27).

Reply: We thank the reviewer for the comment. We agree with the reviewer that the approach to prepare the data set, train and apply the trained model on molecular dynamics, as given in the recent review article¹¹ mentioned by the reviewer and also cited in our manuscript, is general and increasingly routine. However, we argue that there are basically three major novel points of our work.

The first novel point is the application of the trained model to aerosol nucleation studies. The ultimate goal of simulating atmospheric aerosol nucleation is to conduct MD under ambient or laboratory conditions, where the cluster size can easily go beyond the interpolation regime, so how to apply the DNN model with size limited accuracy to atmospheric nucleation becomes a problem that needs to be addressed. This is mostly because the size scalability of the training model is limited around the maximum cluster size in the data set. Here, we determine that collision rate constants can be derived from DNN-based molecular dynamics based on Poisson distribution and embed them into a cluster kinetics model to obtain nucleation rate, bypassing the challenge of training a model whose high accuracy needs to cover clusters with tens of thousands of nucleation molecules.

Second, the novel point is the feasibility of obtaining an accurate reactive force field based on a neural network for aerosol nucleation systems. Aerosol nucleation molecular clusters appear with high flexibility, multicomponent and multiple sizes, posing a huge challenge for obtaining a highly accurate DNN model. There are various existing approaches in each stage of obtaining a DNN model. For instance, for preparing the initial data set, basin-hopping, genetic algorithm, metadynamics and so on are all on the desk, but which one is practically useful for obtaining the accurate DNN model is a question. Additionally, there are quite a few options (active learning, concurrent learning etc.) for obtaining the final data set. Here, we find a practical way to obtain an accurate model for nucleation studies.

Third, the novelty lies in innovation and impact of the whole workflow in the aerosol nucleation simulation field. Notably, there are currently no DNN-based force field application studies in aerosol nucleation simulation field despite the wide popularity of DNN-based force fields in chemistry. In the past decade, multiple groups, including Prof. Hanna Vehkamäki's, Prof. Fangqun Yu's and ours,

published quite a few works (please see the very recent review¹²) investigating aerosol nucleation molecular clusters in a quantum chemical way. The workflow is generally the same: finding the global minimum, calculating energetics and thermodynamics and then conducting cluster kinetics simulation. The workflow we propose here is naturally refreshing in this field. Furthermore, as we emphasize in the title, the workflow will potentially revolutionize the field by making simulations towards fully *ab initio*.

In addition to the lack of novelty, the presented results are not very convincing in my opinion. For example, the DNN-predicted dimer binding curves in Fig. 2a and 2b show strong oscillatory artefacts in the long-range region. Additionally, the correlation plots in 2e, 2f, and 2g show large outliers (several eV/Å) and for many of the clusters, the energy RMSEs presented in 2d are on the order of several kcal/mol. Further, there are some questionable statements. For example, the authors claim that the lack of electrostatic interactions beyond the cutoff does not affect the validity of the proposed framework (line 88 and line 142), even though they admit that the CEF is likely underestimated due to the lack of long-range interactions (line 143). Given that this seems to be crucial, it is unclear to me why the authors did not augment their DNN with long-range electrostatics, in particular because several existing DNNs already do this routinely (see e.g. 10.1021/acs.jctc.9b00181). I would expect that this would greatly improve their results.

Reply: We thank the reviewer for the comments. We agree with the reviewer that long-range interactions should be properly considered, so after obtaining the data set through metadynamics coupled with an active learning strategy in addition to the AIMD trajectory, we switched the short-range DPMD model to the Physnet⁴ model as suggested. In addition, we rechecked the dimer detachment curves and found that the previously used characterization parameter, “binding energy”, was wrongly used. Using binding energy, both the dimers and monomers should be optimized under the same level of method, but we did not optimize the monomers. Then, we looked up the literature¹⁻³ about using machine learning methods to build force fields for molecular clusters and found that typically, “relative energy” instead of “binding energy” is used. Here, relative energy represents the isomer energy minus the energy of the most stable isomer. Therefore we train a deep neural network (DNN) model using Physnet⁴ benchmark the model performance by relative energy, as shown in Fig. 2.

There are also some issues with the structure of the text. To give an example, in the Results section, the authors jump straight to a discussion of passive learning, without any general overview of the individual components of their framework (apart from Fig.1). Even though the methods section provides further details, I think a small paragraph with a general overview would make this section much more readable.

Reply: We thank the reviewer for the suggestion. We agree with the reviewer and added this: *The initial data set was first prepared by metadynamics sampling in addition to subsequent screening and labelling. Then, an active learning strategy with two force thresholds is utilized to supplement the initial data set to form the final data set for training to obtain the DNN-FF. Therefore, multiple nanosecond scale DNN-FF-based MD simulations can be performed. Finally, based on the Poisson distribution, collision rate constants are derived and combined with static quantum chemistry-based*

evaporation rates to obtain macroparameters such as the formation rate by a cluster kinetics model.

Finally, there are many language issues (colloquialism/missing words/wrong grammar), which make the manuscript hard to read. For example (this list is far from exhaustive):

- line 9: "we proposes"

- line 19: "Emergence in 2011 and then broadly employed ..."

- line 25: "However, collision rate constants, derived from simple hard sphere model, are still very rough, which accuracy is far from ..."

- line 44: "... to prepare data set ..."

- line 58: "... that's why ..."

- line 68: "Due to the interpolate nature of DNN ..."

- line 117: "The more inner position ..."

- line 124: "... tons of proton transfer events ..."

- line 155: "... ranges from one to six or so."

I advise additional proofreading, preferably by a native speaker.

Reply: We thank the reviewer for the comments and we are very sorry for these errors. The grammar/syntax/typos errors are corrected as suggested and the manuscript is further polished by author services from Springer Nature.

In summary, I recommend that the authors fix the major flaws and try to submit the paper to a different journal. Even without any of the above-mentioned issues, I don't think this manuscript is suitable for publication in Nature Communications due to the lack of novelty.

Reply: We thank the reviewer for the comment. We revised the manuscript as suggested.

In addition, we run further AIMD calculations and add those structures into the data set to improve the model accuracy. The key issue about the long-range interactions that all reviewers are seriously concerned about is resolved by labelling the data set again with dipole moment information and switching DPMD to Physnet as suggested. Then the retrained model is used to drive MD and subsequently cluster kinetics to obtain the final kinetics which are environmentally relevant and significant.

The novelty of this work lies in proposing a general workflow to practically obtain a highly accurate reactive force field and paving the way to use the training model with limited size scalability for pushing the aerosol nucleation simulation towards fully *ab initio*. The work we propose here not only provides refreshing ideas for the existing nucleation simulation workflows but also presents new insights into the widely recognized nucleation mechanisms that the collision-limited mechanism with zero evaporation rates should be revised to be a collision-enhanced mechanism with small evaporation rates.

Therefore, we believe our work is novel and significant for revolutionizing the atmospheric aerosol nucleation simulation field. Moreover, the work is of broad interest not only to those studying aerosol nucleation but also to the broader molecular simulation community, as suggested by the first reviewer.

Finally, we sincerely thank all the reviewers for their valuable comments, which allowed our work to be greatly improved.

References

- 1 Schran, C., Briec, F. & Marx, D. Transferability of machine learning potentials: Protonated water neural network potential applied to the protonated water hexamer. *J. Chem. Phys.* **154**, 051101 (2021).
- 2 Zhang, L. *et al.* A deep potential model with long-range electrostatic interactions. *J. Chem. Phys.* **156**, 124107 (2022).
- 3 Yao, K., Herr, J. E., Toth, D. W., McKintyre, R. & Parkhill, J. The TensorMol-0.1 model chemistry: a neural network augmented with long-range physics. *Chem. Sci.* **9**, 2261-2269 (2018).
- 4 Unke, O. T. & Meuwly, M. PhysNet: A Neural Network for Predicting Energies, Forces, Dipole Moments, and Partial Charges. *J. Chem. Theory Comput.* **15**, 3678-3693 (2019).
- 5 Yang, M., Bonati, L., Polino, D. & Parrinello, M. Using metadynamics to build neural network potentials for reactive events: the case of urea decomposition in water. *CATALYSIS TODAY* **387**, 143-149 (2022).
- 6 Neese, F., Wennmohs, F., Becker, U. & Riplinger, C. The ORCA quantum chemistry program package. *J. Chem. Phys.* **152**, 224108 (2020).
- 7 Frisch, M. J. *et al.* Gaussian 16 revision a. 03. 2016; gaussian inc. *Wallingford CT* **2** (2016).
- 8 Kulmala, M. *et al.* Direct Observations of Atmospheric Aerosol Nucleation. *Science* **339**, 943-946 (2013).
- 9 Guo, S. *et al.* Remarkable nucleation and growth of ultrafine particles from vehicular exhaust. *Proc. Natl. Acad. Sci. USA* **117**, 3427-3432 (2020).
- 10 Cheng, B. *et al.* Mapping Materials and Molecules. *Acc. Chem. Res.* **53**, 1981-1991 (2020).
- 11 Unke, O. T. *et al.* Machine Learning Force Fields. *Chem. Rev.* **121**, 10142-10186 (2021).
- 12 Elm, J. *et al.* Modeling the formation and growth of atmospheric molecular clusters: A review. *J. Aerosol Sci* **149** (2020).

REVIEWER COMMENTS

Reviewer #1 (Remarks to the Author):

The authors have significantly improved the manuscript. From the point of view of language, the text is much smoother and clearer. From the scientific point of view, the authors have made several improvements:

- The use of a model that explicitly fits long-range electrostatic interactions is definitely the most significant, leading to better reproduction of the interactions.
- They verified that there were discrepancies related to the overlap of DFT and semiempirical relevant structures, thus adding new configurations to the training set.
- They checked the accuracy along the entire reactive process, confirming the goodness of the modeling.

These additions made it possible to better substantiate the arguments proposed in the manuscript, which is thus definitely improved.

I have only a few minor comments:

- Replacing all occurrences of passive learning with metadynamics seems to lead to misunderstandings. In fact, metadynamics is an advanced sampling method that is usually used to sample rare events. As a result, the configurations visited in the space of configurations are greatly increased compared to those sampled using standard molecular dynamics, hence it can be profitably used for training set construction. Instead, by passive learning the authors meant, if I understand correctly, to train a network on a defined dataset of configurations, as opposed to active learning in which the potential is used to generate new configurations to be labeled. However, nothing prevents the use of metadynamics in the active learning phase as well. Consequently, it seems improper to contrast the two. It is suggested to reformulate the various steps in the construction of the training set, first describing them and then stating which method (e.g. metadynamics + furthest point sampling) was used in each phase.
- I think it might be helpful to make explicit why it was changed from using DeepMD to PhysNet. Indeed, in the current form of the manuscript, it is not clear why one potential was used for the training set construction phase and another for the production phase. The reader might ask: why not do everything with PhysNet? Perhaps it might be useful also to keep some of the results obtained with DeepMD in the supporting information.

Reviewer #2 (Remarks to the Author):

With this revised version of the manuscript, the authors addressed many of the reviewers' concerns and solved some of the issues present in the original text. Despite the undeniable efforts put into the revision process, the manuscript still presents some critical issues that must be solved before this work publication.

About the metadynamics part, in the equation (1) the external bias potential is written as a function of the whole set of atomic coordinates R . This is wrong both formally and practically since it has to be a function only of the chosen collective variables. Moreover, given the complexity of the collective variable adopted, equation (1) should come with a reference.

About the range of temperatures investigated, the authors explained why this choice was made, but they did not answer why a wider range was not studied. I believe that with a temperature range of 22 K and with the support of only two points, no claims can be made about the role of temperature in this type of process.

About the MD part, the authors did not provide any result that could prove the validity of their trajectories. The observation of a single reactive event is a clear indication of an incomplete sampling that makes any conclusion derived from such trajectories unreliable. Given the cost-effectiveness of extending DNN-based MD simulations, I do not understand the authors' decision not to further extend their dynamics.

On the subject of MD stability, the high correlation between predicted and calculated energies is not enough to demonstrate the quality of the NNFF. I wonder why the authors did not show conserved

energy and temperature in their simulations. The absence of drift in these two quantities would have been much more solid evidence of their quality and stability.

In conclusion, even though many concerns have been solved after the first step of revision, the presence of many inaccuracies makes this work quality still far from this journal's standards and, regretfully, I can not endorse its publication.

Reviewer #3 (Remarks to the Author):

Although the authors have improved some aspects of the manuscript according to the other reviewers' and my comments, I still believe that this manuscript would be better suited in a specialised journal and does not fit in Nature Communications (see my original review).

In my opinion, applying (by now) standard techniques to a specific field of study is not broadly interesting to an interdisciplinary readership.

Reviewer #1 (Remarks to the Author):

The authors have significantly improved the manuscript. From the point of view of language, the text is much smoother and clearer. From the scientific point of view, the authors have made several improvements:

- The use of a model that explicitly fits long-range electrostatic interactions is definitely the most significant, leading to better reproduction of the interactions.*
- They verified that there were discrepancies related to the overlap of DFT and semiempirical relevant structures, thus adding new configurations to the training set.*
- They checked the accuracy along the entire reactive process, confirming the goodness of the modeling.*

These additions made it possible to better substantiate the arguments proposed in the manuscript, which is thus definitely improved.

Reply: We thank the reviewer for the positive comments.

I have only a few minor comments:

- Replacing all occurrences of passive learning with metadynamics seems to lead to misunderstandings. In fact, metadynamics is an advanced sampling method that is usually used to sample rare events. As a result, the configurations visited in the space of configurations are greatly increased compared to those sampled using standard molecular dynamics, hence it can be profitably used for training set construction. Instead, by passive learning the authors meant, if I understand correctly, to train a network on a defined dataset of configurations, as opposed to active learning in which the potential is used to generate new configurations to be labeled. However, nothing prevents the use of metadynamics in the active learning phase as well. Consequently, it seems improper to contrast the two. It is suggested to reformulate the various steps in the construction of the training set, first describing them and then stating which method (e.g. metadynamics + furthest point sampling) was used in each phase.*

Reply: We thank the reviewer for the suggestions. We agree with the reviewer that comparing metadynamics with active learning is improper, as metadynamics can also be utilized in the active learning iterations. Therefore, we revised the manuscript as requested: *...The initial data set is first prepared by metadynamics sampling in addition to subsequent screening and labelling. The screening is made by farthest point sampling (FPS) while the force and energy labelling is done by density function theory (DFT). Then, an active learning strategy with two force thresholds is utilized to supplement the initial data set to form the final data set to obtain the final force field. In each active learning iteration, DNN-FF-based MD based on the previously active learning iterations selected dataset and metadynamics prepared dataset is conducted. Then, inaccurate structures satisfying the threshold range are selected for labelling and added to the dataset for the next iteration. Therefore, after finalizing the dataset, multiple nanosecond-scale DNN-FF-based MD simulations can be performed. Finally, based on the Poisson distribution, collision rate constants are derived and combined with static quantum chemistry-based evaporation rates to obtain macroparameters such as the formation rate by a cluster kinetics model. The cluster size sampled by metadynamics is based on the cluster*

stability characteristic of acid-base clusters being mostly stable when the difference between acid number and base number is less than or equal to one¹. Active learning not only supplements the structures for metadynamics sample size but also points to the cluster compositions with high evaporation rates, e.g., (DMA)₄, the cluster being composed of four dimethylamine molecules, as we can see from the active learning data set in Fig. 1c, which can normally be ignored through sampling under predefined cluster compositions.

- I think it might be helpful to make explicit why it was changed from using DeepMD to PhysNet. Indeed, in the current form of the manuscript, it is not clear why one potential was used for the training set construction phase and another for the production phase. The reader might ask: why not do everything with PhysNet? Perhaps it might be useful also to keep some of the results obtained with DeepMD in the supporting information.

Reply: We thank the reviewer for the suggestions. The explanation about switching DeePMD to PhysNet is added as requested in the Methods section: *DeePMD with strictly local descriptors is integrated with LAMMPS, which guarantees a high efficiency of MD exploration, so DeePMD is utilized in the active learning iterations. Despite the recent appearance of the long-range version DeePMD (DPLR)², we still switched to Physnet for production, as preparing maximally localized Wannier centers for DPLR requires a large cell where the molecular electron density decreases to zero on the faces of the cell, which is computationally expensive.*

The dimer detachment curves based on the DeePMD model are added to the Supplementary Materials, as shown in Fig. R1. The DeePMD trained model, relative energies of dimers and the script to plot Fig. R1 are all uploaded to figshare (<https://figshare.com/s/ddfa64222a8216c16d36>) for reproducibility.

Fig. R1. Dimer detachment curves for (SA)₁(DMA)₁, (SA)₂ and (DMA)₂, where SA and DMA represent sulfuric acid and dimethylamine molecules, respectively. The relative energy is the isomer energy minus the energy of the most stable isomer. DeePMD represents the model

trained by DeePMD with a cut-off radius of 6.0 Å.

Reviewer #2 (Remarks to the Author):

With this revised version of the manuscript, the authors addressed many of the reviewers' concerns and solved some of the issues present in the original text. Despite the undeniable efforts put into the revision process, the manuscript still presents some critical issues that must be solved before this work publication.

Reply: We thank the reviewer for the comments.

About the metadynamics part, in the equation (1) the external bias potential is written as a function of the whole set of atomic coordinates R . This is wrong both formally and practically since it has to be a function only of the chosen collective variables. Moreover, given the complexity of the collective variable adopted, equation (1) should come with a reference.

Reply: We thank the reviewer for the suggestions. We are deeply sorry for the mistakes in equation (1). We rechecked the equation as well as its descriptions by comparing it with the contents in the original paper³. They have been corrected as shown in the manuscript.

About the range of temperatures investigated, the authors explained why this choice was made, but they did not answer why a wider range was not studied. I believe that with a temperature range of 22 K and with the support of only two points, no claims can be made about the role of temperature in this type of process.

Reply: We thank the reviewer for the comments. We agree with the reviewer that only two points cannot make claims about temperature dependence. Considering that the temperature dependence is a very minor point to the workflow and the two temperatures are the representative temperatures in atmospheric environments, so the claims about temperature dependence are removed. Based on the kinetics under these two temperature, the collision enhancement effects under typical atmospheric conditions can be sufficiently elucidated.

About the MD part, the authors did not provide any result that could prove the validity of their trajectories. The observation of a single reactive event is a clear indication of an incomplete sampling that makes any conclusion derived from such trajectories unreliable. Given the cost-effectiveness of extending DNN-based MD simulations, I do not understand the authors' decision not to further extend their dynamics.

Reply: We thank the reviewer for the comments. There is a significant misunderstanding about the meaning of “reactive event” and we are sorry for not making it clear. The collision rate constants are highly related to the number of reactive events (collision and evaporation of molecules/clusters) we observed in the trajectories. According to the Poisson distribution assumptions⁴, the statistical uncertainty of collision rate constants is decreased when the number of reactive events is increased. Please see more detailed explanations about the

statistical uncertainties of rare events in reactive molecular dynamics simulations in the reference⁴. We did observe many reactive events from the formation of multiple kinds of clusters to the evaporation of unstable clusters (please see the last column in the Supplementary Table 1). This is why we can show collision enhancement factors with uncertainties of these constants below 20 from multiple kinds of collisions, as shown in Fig. 4a and in Supplementary Table 1. The single reactive event that the review mentioned is from the original Supplementary Figure 4. Notably, the time evolution of the proton number shown in Supplementary Figure 4 is the period after the formation of (SA)₆(DMA)₆, which is from the collision of (SA)₂(DMA)₁ and (SA)₄(DMA)₅. It is only one part of the whole one nanosecond trajectory. For the thorough sampling of clusters' potential energy surface, it is clearly out of the focus, as we intend to run the MD simulation to study the molecules nucleation processes instead of finding the global minimum. Here is the reason why we did not further extend the trajectory to go beyond one nanosecond. At one nanosecond, two larger clusters are formed (Table R1), close to equilibrium, so further extending the trajectory can provide little information about kinetics with small uncertainties and only give the kinetics of large clusters with a large uncertainty since the collisions among large clusters are rare, which is basically useless for further cluster kinetics simulations due to the larger uncertainty.

temperature	trajectory #	Cluster composition at 1 ns
278 K	0	(SA) ₆ (DMA) ₆ , (SA) ₄ (DMA) ₄
	1	(SA) ₅ (DMA) ₄ , (SA) ₅ (DMA) ₆
	2	(SA) ₅ (DMA) ₆ , (SA) ₅ (DMA) ₄
300 K	0	(SA) ₇ (DMA) ₆ , (SA) ₃ (DMA) ₄
	1	(SA) ₆ (DMA) ₆ , (SA) ₄ (DMA) ₄
	2	(SA) ₅ (DMA) ₃ , (SA) ₅ (DMA) ₇

Table R1. The cluster composition at 1 ns for each trajectory conducted at 278 K and 300 K.

On the subject of MD stability, the high correlation between predicted and calculated energies is not enough to demonstrate the quality of the NNFF. I wonder why the authors did not show conserved energy and temperature in their simulations. The absence of drift in these two quantities would have been much more solid evidence of their quality and stability.

Reply: We thank the reviewer for the suggestions. We agree with the reviewer that the conserved energy and temperature can provide additional evidence to validate the stability of MD. Therefore, we add the model performance as shown in Fig. R2. Notably, verifying the energy conservation of MD simulations is typically conducted under NVE ensemble⁵, so we benchmarked the MD stability under NVE ensemble. The total energy remains stable with a change of approximately 0.1 eV. Due to the heat from molecular collisions, the temperature under the NVE ensemble keeps growing gradually. This information has also been added to the Supplementary Materials.

Fig. R2. Time dependences of the total energy (the relative value to the first snapshot) and temperature during the MD simulation under the NVE ensemble.

In conclusion, even though many concerns have been solved after the first step of revision, the presence of many inaccuracies makes this work quality still far from this journal's standards and, regrettably, I can not endorse its publication.

Reply: We thank the reviewer again for the comments that could help us further improve our work.

Reviewer #3 (Remarks to the Author):

Although the authors have improved some aspects of the manuscript according to the other reviewers' and my comments, I still believe that this manuscript would be better suited in a specialised journal and does not fit in Nature Communications (see my original review).

In my opinion, applying (by now) standard techniques to a specific field of study is not broadly interesting to an interdisciplinary readership.

Reply: We thank the reviewer for the comments. In retrospect, we think that the most valuable idea in the workflow is to derive collision rate coefficients from DNN-FF-based MD. This idea not only utilizes the high accuracy of DNN-FF but only bypasses the cluster size generalization ability problem. Therefore, the workflow is not a standard technique but a new way to solve

problems in atmospheric chemistry.

In addition, we believe that the workflow is broadly interesting. For those who are interested in atmospheric chemistry, we provide the workflow to revisit the widely recognized nucleation mechanism in a refreshing way, in contrast to the typical calculation protocols applied in aerosol nucleation simulation⁶. For those who are interested in molecular simulation, or even machine learning, fully simulating aerosol nucleation at the accuracy of first principles calls for new machine learning tricks. For instance, enlarging the cluster size in the dataset calls for a more compact dataset, where recently proposed E(3)-equivariant graph neural networks⁷ could help. Improving the accuracy from density function theory to wave function theory might benefit from transfer learning or delta learning. Notably, new machine learning techniques should be integrated in the workflow we proposed because as far as we know, there is no report that integrates machine learning force fields into aerosol nucleation simulations field except one very recent study⁸ which only intends to speed up the screening process in the well-established nucleation calculations workflow⁶. Therefore, we believe that our work is of broad interest to those who are interested in atmospheric chemistry, molecular simulation and machine learning.

References

- 1 Almeida, J. *et al.* Molecular understanding of sulphuric acid-amine particle nucleation in the atmosphere. *Nature* **502**, 359-363 (2013).
- 2 Zhang, L. *et al.* A deep potential model with long-range electrostatic interactions. *J. Chem. Phys.* **156**, 124107 (2022).
- 3 Herr, J. E., Yao, K., McIntyre, R., Toth, D. W. & Parkhill, J. Metadynamics for training neural network model chemistries: A competitive assessment. *J. Chem. Phys.* **148**, 241710 (2018).
- 4 Kröger, L. C., Kopp, W. A., Döntgen, M. & Leonhard, K. Assessing Statistical Uncertainties of Rare Events in Reactive Molecular Dynamics Simulations. *J. Chem. Theory Comput.* **13**, 3955-3960 (2017).
- 5 Zeng, J., Cao, L., Xu, M., Zhu, T. & Zhang, J. Z. H. Complex reaction processes in combustion unraveled by neural network-based molecular dynamics simulation. *Nat. Commun.* **11**, 5713-5713 (2020).
- 6 Elm, J. *et al.* Modeling the formation and growth of atmospheric molecular clusters: A review. *journal of aerosol science* **149** (2020).
- 7 Batzner, S. *et al.* E(3)-equivariant graph neural networks for data-efficient and accurate interatomic potentials. *Nat. Commun.* **13**, 2453 (2022).
- 8 Kubečka, J., Christensen, A. S., Rasmussen, F. R. & Elm, J. Quantum Machine Learning Approach for Studying Atmospheric Cluster Formation. *Environmental Science & Technology Letters* **9**, 239-244 (2022).

REVIEWERS' COMMENTS

Reviewer #1 (Remarks to the Author):

The authors, having already made numerous changes to the manuscript in the previous revision stage, have corrected other minor aspects as well. The methodology is now more convincing and appropriate to the problem studied. Thus I think it is now suitable, at least from the point of view of scientific soundness, for publication.

Reviewer #2 (Remarks to the Author):

Even though the authors clarified many of the reviewers' claims, I found the manuscript hard to follow in many parts and its readability still poor.

The whole procedure is a Chimera of methods roughly introduced and merged together that makes this paper not easily accessible for a broad audience. Moreover, the use of more than one software for similar tasks (Gaussian and Orca for electronic structure calculations, DeepMD and PhysNet for DNN-FF training) makes the whole process not elegant and more suited for applications research than that for presenting a general workflow. Simpler and more transparent methods are already present in the literature and the one introduced in this work did not prove to perform any better than the others.

A few comments:

At line 60 the authors claim that "multiple nanosecond-scale DNN-FF-based MD simulations can be performed". However, they never did simulations longer than a single nanosecond.

However no restraints are applied on the temperature, this quantity cannot drift forever and in fig. R2 this time series had to be shown until a plateau was reached. An endless drift may indicate an insufficient equilibration or the presence of severe issues in the interatomic potential. Why did the authors not show the behaviour of a longer simulation? Are these simulations stable even in the absence of a thermostat, or were they truncated because unstable after a few hundreds of picoseconds?

Because of all these reasons, I cannot change my previous decision and I do not recommend this paper for publication.

Reviewer #1 (Remarks to the Author):

The authors, having already made numerous changes to the manuscript in the previous revision stage, have corrected other minor aspects as well. The methodology is now more convincing and appropriate to the problem studied. Thus I think it is now suitable, at least from the point of view of scientific soundness, for publication.

Reply: We thank the reviewer for the positive comments.

Reviewer #2 (Remarks to the Author):

Even though the authors clarified many of the reviewers' claims, I found the manuscript hard to follow in many parts and its readability still poor.

The whole procedure is a Chimera of methods roughly introduced and merged together that makes this paper not easily accessible for a broad audience. Moreover, the use of more than one software for similar tasks (Gaussian and Orca for electronic structure calculations, DeepMD and PhysNet for DNN-FF training) makes the whole process not elegant and more suited for applications research than that for presenting a general workflow. Simpler and more transparent methods are already present in the literature and the one introduced in this work did not prove to perform any better than the others.

Reply: We thank the reviewer for the comments. The workflow we introduced here is benchmarked and further validated by deriving atmospherically relevant kinetics parameters based on a deep learning force field. Furthermore, each module in the workflow has been described in detail, and the related codes, scripts and the input and output files are all uploaded in a public repository so that they are easily accessible for reproduction. The workflow is general on how to simulate atmospheric aerosol nucleation fully *ab initio*. The software packages such as the electronic structure calculation packages and deep neural network force field training packages, utilized in each module of the workflow, can be changed as long as their output information is the same as that requested in the workflow. Therefore, the specific packages are not as important as the logic itself in the workflow. To the best of our knowledge, the general workflow towards fully *ab initio* simulation of aerosol nucleation does not yet exist except for the work we proposed here.

A few comments:

At line 60 the authors claim that "multiple nanosecond-scale DNN-FF-based MD simulations can be performed". However, they never did simulations longer than a single nanosecond.

Reply: We thank the reviewer for the comment. We revised the claim to “multiple DNN-FF-based one nanosecond MD simulations can be performed” to clarify the duration of MD simulations. The main reason for not simulating the dynamics longer than one nanosecond is explained before: at one nanosecond, two larger clusters are formed, close to equilibrium, so

further extending the trajectory can provide little information about kinetics and only give the kinetics of large clusters with a large uncertainty since the collisions among large clusters are rare, which barely contributes to the further cluster kinetics simulations. The minor reason is that after switching the model for the final force field training from DeePMD to Physnet, the MD efficiency of ASE, which is coupled with Physnet, is much lower than that of LAMMPS, which is coupled with DeePMD, making us reluctant to extend the simulation unless it is necessary.

However no restraints are applied on the temperature, this quantity cannot drift forever and in fig. R2 this time series had to be shown until a plateau was reached. An endless drift may indicate an insufficient equilibration or the presence of severe issues in the interatomic potential. Why did the authors not show the behaviour of a longer simulation? Are these simulations stable even in the absence of a thermostat, or were they truncated because unstable after a few hundreds of picoseconds?

Reply: We thank the reviewer for the comment. We agree with the reviewer that the temperature will not drift infinitely and will stop drifting when equilibration is reached. However, reaching complete equilibration means that only one large cluster is formed, which is very time-consuming despite the relatively high efficiency of the deep neural network-based force field. Instead, we extend the original 100 ps to 200 ps under the NVE ensemble, and we expect that the steady temperature duration will be longer; in other words, the plateau duration will be longer. This is precisely what we saw as shown below, indicating the high stability of MD simulations.

Fig. R1. Time dependences of the total energy (the relative value to the first snapshot) and temperature during the MD simulation under the NVE ensemble.

Because of all these reasons, I cannot change my previous decision and I do not recommend this paper for publication.

Reply: We thank the reviewer for the comments. The MD simulation duration issue is mainly because at the end of one nanosecond, two larger clusters are formed, so further extending the trajectory can provide little information about collision kinetics, which barely contributes to the further cluster kinetics simulations. For the temperature drift issue, the plateau can be more clearly seen after we conducted MD simulations over a longer period. In summary, we believe that those two important issues are well resolved, and we sincerely hope that the reviewer can reconsider our work for publication.